# Giant sponge grounds of Central Arctic seamounts are associated with extinct seep life

T. M. Morganti [1✉], B. M. Slaby [2], A. de Kluijver [3], K. Busch [2], U. Hentschel [2,4], J. J. Middelburg [3,5], H. Grotheer [6], G. Mollenhauer [5,6], J. Dannheim [6,7], H. T. Rapp[8,9], A. Purser [6] & A. Boetius [1,5,6✉]

The Central Arctic Ocean is one of the most oligotrophic oceans on Earth because of its sea-ice cover and short productive season. Nonetheless, across the peaks of extinct volcanic seamounts of the Langseth Ridge (87°N, 61°E), we observe a surprisingly dense benthic biomass. Bacteriosponges are the most abundant fauna within this community, with a mass of 460 g C m$^{-2}$ and an estimated carbon demand of around 110 g C m$^{-2}$ yr$^{-1}$, despite export fluxes from regional primary productivity only sufficient to provide <1% of this required carbon. Observed sponge distribution, bulk and compound-specific isotope data of fatty acids suggest that the sponge microbiome taps into refractory dissolved and particulate organic matter, including remnants of an extinct seep community. The metabolic profile of bacteriosponge fatty acids and expressed genes indicate that autotrophic symbionts contribute significantly to carbon assimilation. We suggest that this hotspot ecosystem is unique to the Central Arctic and associated with extinct seep biota, once fueled by degassing of the volcanic mounts.

[1] Max Planck Institute for Marine Microbiology, Celsiusstr. 1, 28359 Bremen, Germany. [2] GEOMAR Helmholtz Centre for Ocean Research Kiel, Düsternbrooker Weg 20, 24105 Kiel, Germany. [3] Utrecht University, Department of Earth Sciences, Princetonlaan 8a, 3584 CB Utrecht, The Netherlands. [4] Christian-Albrechts University of Kiel, Christian-Albrechts-Platz 4, 24118 Kiel, Germany. [5] MARUM and Department of Geosciences, University of Bremen, 28359 Bremen, Germany. [6] Alfred Wegener Institute Helmholtz Center for Polar and Marine Research, Am Handelshafen 12, 27570 Bremerhaven, Germany. [7] Helmholtz Institute for Functional Marine Biodiversity, Ammerländer Heerstraße 231, 26129 Oldenburg, Germany. [8] University of Bergen, Department of Biological Sciences and K.G. Jebsen Centre for Deep-Sea Research, PO Box 7803, 5020 Bergen, Norway. [9]Deceased: H. T. Rapp. ✉email: tmorgant@mpi-bremen.de; antje.boetius@awi.de

Dense aggregations of large sponges, typically known as sponge gardens or grounds[1], have been reported from intertidal to abyssal depths[1]. As with coral reefs, sponge grounds are characterized by a progressive accumulation of biomass produced by a limited number of long-lived benthic animal species. The sponges act as ecosystem engineers: they influence community structure by providing distinct habitat niches, such as a suitable substrate for organisms to settle on, hide within or prey from, thereby enhancing local biodiversity[1]. Via their utilization of suspended material, sponges often also play an important role in nutrient and organic matter cycling between the benthic and pelagic realms[1,2].

Sponges are primarily filter feeders, with the densest communities often occurring where hydrodynamic conditions favor the localized concentration of particulate organic carbon[3–5], such as at the interface between two water masses[6,7]. Many of the densest deep-sea sponge grounds have been found on the shelf breaks of the North Atlantic, the Western Canadian margin, the slopes of the Hawaiian archipelago, in the Antarctic, and the Mediterranean Sea[1]. Sponge abundances in these areas range from 0.01 to 25 individuals $m^{-2}$ [3,5], with wet weights (WW) for these sponge communities of up to 30 kg $m^{-2}$ [8]. In such dense aggregations, sponge grounds can filter hundreds of liters of water[9]. This filtering influences particle abundance within the overlying waters[10], nutrient cycling[2], and has significant consequences for local biogeochemical cycles and food webs[2,11].

Sponges exploit different food sources; in addition to particulate organic matter (POM), they also effectively remove dissolved organic matter (DOM) from seawater[10,12]. Sponge grounds of Arctic-Boreal regions are often dominated by large high-microbial abundance species (HMAs, i.e., sponge species which host highly diverse and dense microbial communities in their tissue) of the genus *Geodia*, co-occurring with *Stelletta* and *Thenea*[13]. Such communities have been observed at the Faroes[4], on the North Norwegian shelf[11], and within the North-West Atlantic[3], with individuals attaining diameters of up to 1 m and WW of 25 kg[4]. *Geodia* spp. host highly diverse microbial communities that can reach abundances of $10^8$ microbes $g^{-1}$ of sponge tissue[14], contributing significantly to the microbial diversity within deep-sea ecosystems[15]. Associated microorganisms contribute to the health and nutrition of the sponges by producing antibiotics, transferring nutrients to the host, and processing sponge-metabolic waste[16].

In this study, we investigate the surprisingly sponge-rich benthic community of Langseth Ridge (87°N, 61°E). This ridge forms part of the Eastern Volcanic province of the ultraslow-spreading Gakkel Ridge and is comprised of a chain of three currently hydrothermally inactive seamounts. These mounts run north-south from an intersection with the Gakkel Ridge, with the Northern and Central mounts smaller than the most southerly Karasik seamount[17,18]. This section of the Gakkel Ridge is ice-covered throughout the year, characterized by very low pelagic and sea-ice algae primary productivity (<25 g C $m^{-2}$ $yr^{-1}$) and low export rates (<1 g C $m^{-2}$ $yr^{-1}$)[19,20], barely sufficient to meet the carbon demand of typical Arctic Ocean assemblages of planktonic and benthic life[21]. To identify the food sources supporting the rich sponge community occupying these seamount summits, we combine under-ice seafloor mapping with biomass sampling to assess the bulk and compound-specific carbon (δ$^{13}$C and Δ$^{14}$C) and nitrogen (δ$^{15}$N) isotopes and fatty acid (FA) composition of sponge tissue and associat this community, as well as the isotope ratios of putative suspended and sedimented particulate food sources. In addition, omics techniques are used to assess the sponge microbiome composition and the expressed functional genes. In the absence of active hydrothermal venting at the investigated seamounts, we hypothesize that the sponges use the refractory organic matter (OM) trapped in the spicule-tube mat on which they sit, as carbon, nitrogen, and energy source. A substantial proportion of such refractory OM is the remnant material of past seep biota, apparently produced during a phase of active venting of the seamounts several thousand years ago.

## Results

**Sponge community description, density, and biomass.** Geological sampling indicated that the seamounts are formed by basalt rock with brecciated rock inclusions of volcanic rock clasts[22]. The Ocean Floor Observation and Bathymetry System (OFOBS) survey data showed that the seamount summits (721–585 m) were covered by massive sponges[22] representing the densest Arctic sponge ground found to date (Figs. 1, 2).

The bulk of the visible sponge biomass comprised individuals from the HMA species *Geodia parva*, *G. hentscheli*, and *Stelletta rhaphidiophora* (Class Demospongiae), resembling the community composition of a typical Arctic-Boreal *Geodia* ground as previously reported from 150–1700 m depths and latitudes of 40°–75°N[1]. Although less abundant, glass and calcareous sponges of several distinct species were also observed, including the discovery of a new genus and species *Sarsinella karasikensis*[23]. The majority of individual sponges were colonized by bryozoan colonies primarily belonging to one family, cf. Crisiidae, and several species of tube-dwelling polychaetes, mostly Serpulidae. These serpulid tubes were affixed within the sponge spicules and grew radially from the equatorial area of each sponge (Fig. 1d). Various species of glass sponges were also observed growing directly on the massive Demosponges. Shrimps were relatively abundant at densities of about 1 ind. $m^{-2}$, as were asteroids and ophiuroids at lower densities (average density ‹0.2 ind. $m^{-2}$), though still at densities significantly higher than observed on the seafloor surrounding the seamounts[24]. To the best of our knowledge, octocoral colonies were observed growing on sponges for the first time in these high latitudes (Fig. 1d).

The dense sponge grounds covered an area of >15 km², across the peaks and saddles of the Langseth Ridge seamount chain (Fig. 1a). The densest sponge aggregations (from 7 to 11 ind. $m^{-2}$) were observed across the flat upper regions of each peak (721–585 m), an area of ~2.5 km² (Figs. 1b, c, 2). In these areas, the siliceous spicules produced by sponges formed a dense mat underlying the living community[22] (Fig. 1e and Supplementary Fig. 1a, b), often intermixed with thick layers of empty siboglinid and serpulid polychaete tubes and bivalve shells (hereafter referred to as "spicule-tube mat", Fig. 1e–g and Supplementary Fig. 1). The most frequent empty worm tubes observed within this mat were identified as those of *Polybrachia* (Family: Siboglinidae; Frenulata) (Fig. 1f). These worms commonly associate with vents and seeps, host chemoautotrophic bacteria, and form tubes from chitin and proteinaceous matter[25,26]. However, no living siboglinid colony was found during our survey. The average OM content of the spicule-tube mat structure was $8 ± 1\%$ of DW. Shells of an unknown blackened mytilid (not further identified) were also abundant within the spicule-tube mat. Empty bivalve shells from the species *Limatula hyperborea* and *Portlandia arctica*, were also frequently found within the mat, species which are typically found throughout Arctic shelf seas and previously found across summits of seamounts at greater depths[27]. A detailed list of identified bivalve shells is given in Supplementary Table 1.

Seafloor coverage by sponges followed a depth-dependent abundance gradient (Fig. 2). Across the flat summits of the Langseth Ridge seamounts, numerous sponges were observed on top of a thick mat comprised of primarily sponge spicules (category c, depth ~700–750 m to ~950–1000 m, average sponge

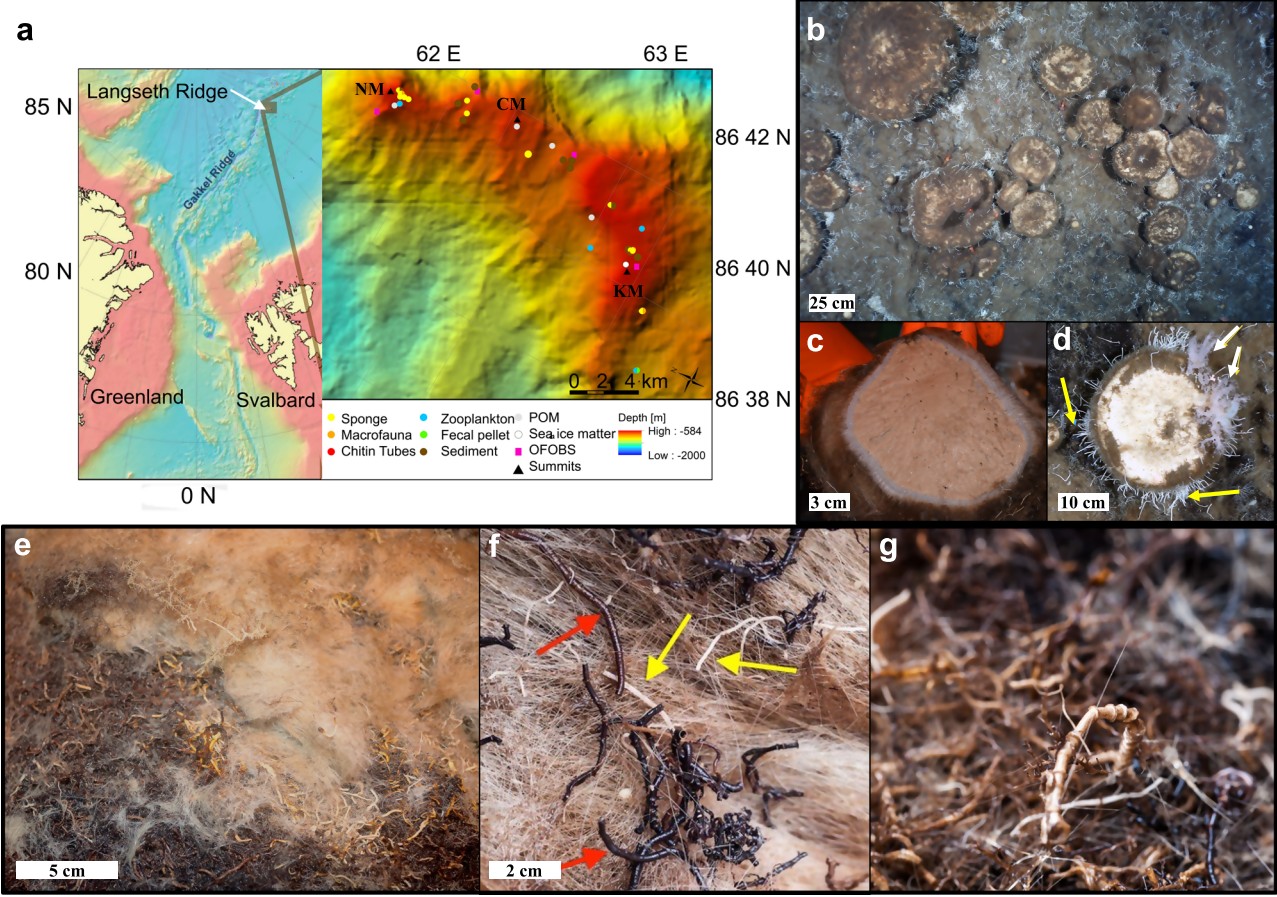

**Fig. 1 Sponge ground landscape and features.** **a** Map showing the study area in the Central Arctic and sample types collected at different sampling stations along the Langseth Ridge (NM Northern mount, CM central mount, KM Karasik seamount); **b** Arctic *Geodia* sponge ground (photo credits PS101 AWI OFOBS); **c** *Geodia* sp. profile (photo credits Jennifer Dannheim); **d** Details of a sponge specimen colonized by several species of tube-dwelling polychaetes arranged radially from the equatorial area of the sponge (yellow arrows), and soft corals growing on a sponge (white arrows) (photo credits PS101 AWI OFOBS); **e** Detailed spicule-mat like structure: the light beige part is composed of sponge spicules, the dark brown part is composed of empty worm tubes (photo credits Beate Slaby); **f** A detailed picture of the empty siboglinid (red arrows) and serpulid (yellow arrows) tubes (photo credits Mario Hoppmann); and **g** A detailed picture of the intricate matrix of empty tubes (photo credits Antje Boetius).

density ±SD: 2.8 ± 1.1 ind. m$^{-2}$, $n = 211$ images, Fig. 2c). The percentage of seafloor covered by living sponges was particularly high within the highest central region of these flat peaks, with >50% of the seafloor covered by living sponges (category d, depth ~700–750 m and shallower, 5.9 ± 1.7 ind. m$^{-2}$, $n = 54$ images, Fig. 2d). Here, the mat-like structure of sponge spicules was up to 15 cm thick, overlying partially blackened siboglinid tubes (Fig. 1e–g and Supplementary Fig. 1a, b). Surrounding the flat central summit areas, toward the steeper edges of the ridge complex, the blackened siboglinid tube detritus was more regularly exposed, with this detritus only occasionally overgrown by sponges or covered with spicules (category b, depth ~950−1000 m to ~1200 m, 0.8 ± 0.7 ind. m$^{-2}$, $n = 361$ images, Fig. 2b). On the steep flanks of the Langseth Ridge, flat or low angle shelf outcrops to depths of ~1300 m supported smaller sponge assemblages of <100 individuals per shelf (category a, Fig. 2a). Occasional stalked glass sponges were observed at depths of 750–1050 m on steeper (>10° to near-vertical) walls. Very few sponges were observed on the deep basalt flanks of the seamounts or within the Gakkel Ridge rift valley (1200–3500 m depth)[22].

Individuals of the three dominant Demospongiae species (*G. parva*, *G. hentscheli*, and *S. rhaphidiophora*) had a median diameter of 17 cm (range 1.5–110 cm, SE = 0.10 cm, $n = 10,839$), with the lower end of this range representing juvenile sponges

(Supplementary Fig. 2, inter-quartile ranges 11–25 cm). Sponge biomass was estimated at 21.9 ± 12.5 kg WW m$^{-2}$ (4.1 ± 2.3 kg DW m$^{-2}$, average ± SD, inferred from literature conversion factor[11,28]) when averaged over the sponge abundant areas of the three summits (the regions of c and d categories of sponge coverage). Biomass was highest on the Northern Mount (NM), estimated to be 66 kg WW m$^{-2}$. The organic carbon content in sponges was 30 ± 5% of DW, i.e., on average 1213 ± 690 g C$_{org}$ m$^{-2}$ from the most densely populated areas (category c and d) and 456 ± 190 g C$_{org}$ m$^{-2}$ for from across the entire surveyed area.

Many sponges showed substantial budding (Supplementary Fig. 3a), indicating active reproduction occurring within the community. The small juveniles, of <1 cm diameter, appear to separate from the adult sponges and were observed almost exclusively in direct proximity of adults (Supplementary Fig. 3b, c), with an average abundance of 3.4 ± 1.5 small juveniles m$^{-2}$, and a maximum observed abundance of 29.3 small juveniles m$^{-2}$ at the NM.

**Stable carbon and nitrogen isotope value.** Stable carbon and nitrogen isotope ratios of demosponge tissue ranged from −19.3 to −18.2‰ and from 6.6 to 8.4‰, respectively (Figs. 3a, 4 and Table 1). The δ$^{13}$C and δ$^{15}$N values were significantly different between adult sponges and, as well as between sponge species

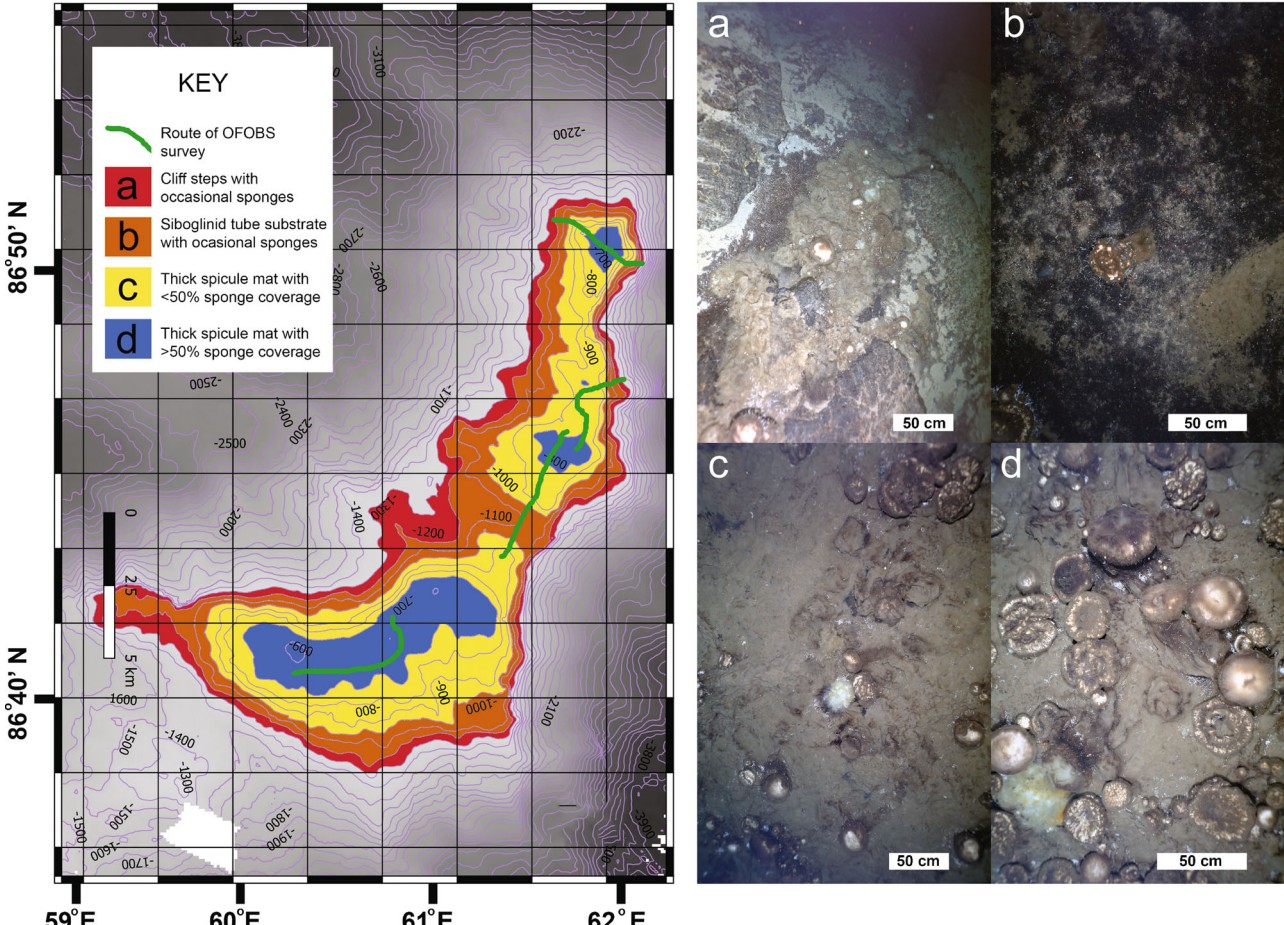

**Fig. 2 Sponge density map and representative seafloor images. (Map)** Sponge abundances estimated for different habitat categories: category "a", on flat shelves on the steep flanks of the Langseth Ridge (red, occasional concentrations of <100 individuals); category "b", on shallow sloping upper flanks, with abundant blackened siboglinid tube detritus exposed (orange, average ± SD: 0.8 ± 0.7 ind. m$^{-2}$, 202 ± 195 g C$_{org}$ m$^{-2}$); category "c", on the flat summits predominantly covered by sponge spicule-mat with numerous sponges, also present (yellow, average ± SD: 2.8 ± 1.1 ind. m$^{-2}$, 725 ± 217 g C$_{org}$ m$^{-2}$); and category "d", on the center of flat seamount peaks (blue, average ± SD: 5.9 ± 1.7 ind. m$^{-2}$, 1702 ± 904 g C$_{org}$ m$^{-2}$). OFOBS dive tracks are shown in green. (Seafloor images **a**–**d**) white scale bars 50 cm length: **a** Category "a": at the edge of the flat summit, a rapid increase in slope correlates with a sharp decrease in sponge abundance. Sand and rock outcrops are covered with the siboglinid tube mat. **b** Category "b": on the more gently shallowing slopes of the mount peaks, dense sponge cover is replaced by occasional individuals on top of the siboglinid tube mat. **c** Category "c": dense agglomerations of sponges cover the flat summits, with occasional gaps in sponge spicule cover visible and with 100% tube cover within many of these. **d** Category "d": in the shallow center of flat summits, where sponge abundances exceed >50% of seafloor coverage. (Photo credits PS101 AWI OFOS system).

$(F_{(3,30)} = 8.32, \quad p = 0.0005 \quad$ and $\quad F_{(3,30)} = 10.48, \quad p = 0.0001,$ respectively). A post hoc Tukey test indicated that *G. parva* juveniles had a significantly lower δ$^{15}$N ratio than adult sponges. Similarly, the δ$^{13}$C ratio significantly differed between *G. parva* juveniles and adults, as well as between *G. parva* and *G. hentscheli* adults (post hoc Tukey test $p < 0.01$, Supplementary Table 2a and Supplementary Fig. 4). Sample location did not show a significant effect ($p > 0.2$) on isotopic values.

Although bulk isotope values differed among sponge species, the variability between sponge tissue (all species pooled together) and other sample types (sediment, macrofauna, POM, and fecal pellet) was greater (Figs. 3a, 4 and Table 1). Post hoc Tukey comparisons indicated that the mean δ$^{15}$N of sponge tissue differed significantly from all sample types, including all macrofauna, with the exception of siboglinid tubes and zooplankton. Similarly, the mean δ$^{13}$C of sponge tissue differed significantly from all sample types with the exception of siboglinid tubes and sponge-associated macrofauna (post hoc Tukey comparisons $p > 0.05$, Supplementary Table 2b). The sampled calcareous sponge had a more depleted δ$^{13}$C value than

the Demospongiae samples, with a value more similar to that of suspended POM and fecal pellets (Fig. 4 and Table 1).

**Radiocarbon Δ$^{14}$C.** The comparison of carbon isotopes δ$^{13}$C and Δ$^{14}$C using both published data and results from the present study show that the Δ$^{14}$C of sponge bulk biomass was most similar to that of the bacterial FA (*i*C15:0) and to that of the bryozoans growing on the sponge, as well as being within the range reported for dissolved inorganic carbon (DIC)[29] in the Arctic at similar depths (Fig. 3b). The Δ$^{14}$C of OM in the siboglinid tubes and sedimentary material, as well as for the locally measured DOC[29] were lower than that of sponge bulk biomass. This suggests that the bulk of the carbon incorporated into the sponges was partially derived from DIC; most likely via autotrophy of their symbionts. Figure 3C shows that according to Δ$^{14}$C and δ$^{15}$N values, detrital material from the siboglinid tubes likely also plays an important role in nutrition.

The age of the small juvenile sponges collected (1–5 cm diameter) was measured to be quite uniform (133 ± 7.2 annum). In the large

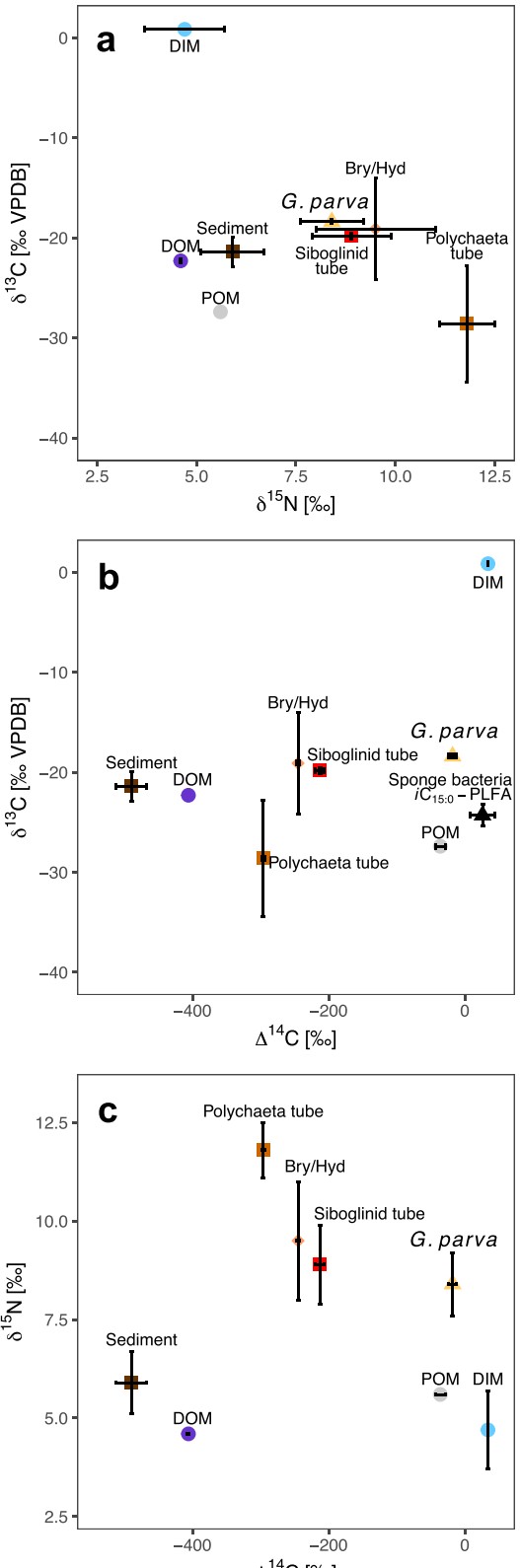

**Fig. 3 Carbon and nitrogen isotope bi-plots.** Isotope bi-plots using **a** $\delta^{13}C$ vs. $\delta^{15}N$; **b** $\delta^{13}C$ vs. $\Delta^{14}C$; and **c** $\delta^{15}N$ vs. $\Delta^{14}C$ (if repeat measurements of a sample was possible data shown represents mean ± SD, otherwise, the error bar indicates 1σ analytical uncertainty. Sample size is provided in Table 1 for each sample type). Sponges (*G. parva*, yellow triangle), bryozoan and hydrozoan skeletons (Bry/Hyd, orange diamond), polychaeta tubes (light brown square), siboglinid tubes (red square), sponge-bacteria FA *i*C15:0-PLFA (black triangle) and sediment (brown square). Dissolved inorganic matter (DIM, light blue circle), particulate organic matter (POM, light gray circle), and dissolved organic matter (DOM, purple circle) values are from the literature[29,93,104,105]. The DIM values refer to dissolved inorganic carbon (DIC) and nitrate. The inorganic $\Delta^{14}C$ was measured in calcareous polychaeta tubes and bryozoan and hydrozoan skeletons. Note that the $\Delta^{14}C$-DIC carries a post-bomb signature in the top 1000 m water depth[29]. Source data are provided as a Source Data file.

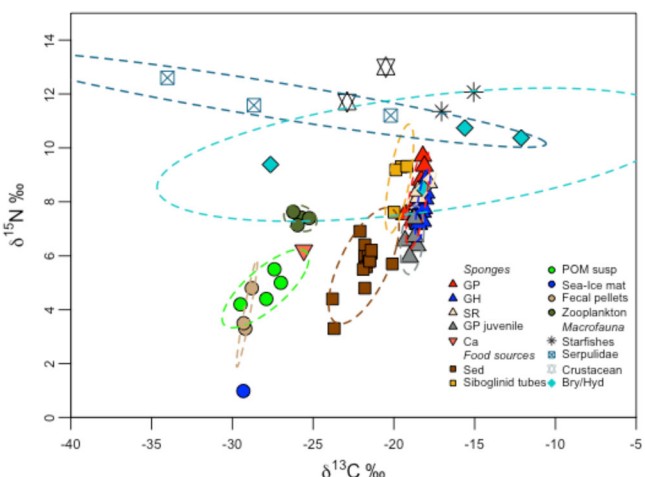

**Fig. 4 Bulk stable N and C isotopes of Karasik benthic community.** Stable isotope $\delta^{13}C$ and $\delta^{15}N$ values (‰) for sponges, different organic matter sources, and associated macrofauna. Standard ellipse areas (SEAc) depicted by dashed lines provide estimates of the niche area of each of these groups using Bayesian inference as in[106]. For crustaceans, asteroids, sea-ice matter, and calcareous sponge (Ca) the SEAc was not calculated due to limited sample size ($n < 3$). Source data are provided as a Source Data file. GP *G. parva*, GH *G. hentscheli*, SR *S. rhaphidiophora*, GP juvenile *G. parva* juvenile, Ca calcareous sponge, sed sediments, POM susp suspended particulate organic matter, Sea-Ice-mat sea-ice matter, Bry/Hyd bryozoans/hydrozoans.

sponge specimen, tissue age increased with radial distance from sponge center; the outer layer (below the cortex) and inner sections of tissue (separated by 9 cm) differed by 163 a, indicating a growth rate of 0.55 mm per year. Thus, we conclude an average sponge age of ca. 300 years for the average adult sponges of the Langseth Ridge seamount community. The ages of all material collected from and comprising the underlying mats, i.e., siboglinid and calcareous

serpulids tubes, and bryozoans, indicated this material to be much older (2392 ± 449 a) than the sponge tissue. The radiocarbon age of sediment samples collected from beneath the spicule-tube mat was found to be considerably older than sediments directly exposed to the water column, by >1400 a. Bivalve shells represented the oldest material from within the underlying hydrothermal community material (7162 ± 29 a) (Table 1).

**PLFAs composition and CSIA.** Bacteria-, algae- and sponge-specific FAs were found in all sponge species, contributing differently to the total phospholipid-derived fatty acids (PLFAs). Bacteria PLFAs were the most abundant (>60%), among which 9-me-C16:0 and 11-me-C18:0 (hereafter grouped as mid-methyl-branched, MBFA $C_{16}$–$C_{18}$) represented 22–32% of total PLFAs. Monounsaturated FAs (MUFA) C16:1ω7 and C18:1ω7 were also abundant among bacterial markers, generally representing 16–18% of total PLFAs. Sponge PLFAs ($\geq C_{24}$) such as Me-$C_{24:2}\Delta^{5,9}$ and $C_{26:2}\Delta^{5,9}$ contributed less to total PLFAs,

**Table 1 Stable and radiocarbon isotope values.**

| Summits | | 13 C (‰) | | 15 N (‰) | | n | Δ14 C (‰) | | *n* |
|---|---|---|---|---|---|---|---|---|---|
| | | mean | sd | mean | sd | | mean | 1σ | |
| **Sponges** | **Species** | | | | | | | | |
| KM | *G. parva* | −18.40 | 0.15 | 8.43 | 0.78 | 7 | −16 (200 a) | 6 | 1 |
| | *G. hentscheli* | −18.19 | 0.22 | 8.16 | 0.51 | 4 | | | |
| | *S. rhaphidiophora* | −18.28 | 0.36 | 8.42 | 0.28 | 3 | | | |
| | *G. parva* juveniles | −18.87 | 0.30 | 6.59 | 0.52 | 5 | | | |
| CM | *G. parva* | −19.33 | 0.82 | 7.52 | 0.04 | 1 | | | |
| | *G. hentscheli* | −18.16 | 0.31 | 7.75 | 0.50 | 1 | | | |
| | *S. rhaphidiophora* | − | − | − | − | − | | | |
| NM | *G. parva* | −18.62 | 0.15 | 7.72 | 0.28 | 5 | −9 (133 a) | 6 | 1 |
| | *G. hentscheli* | −18.26 | 0.22 | 7.52 | 0.51 | 3 | | | |
| | *S. rhaphidiophora* | − | − | − | − | − | | | |
| | Calcareous sponge | −25.58 | | 6.20 | | 1 | | | |
| CS | *G. parva* | −18.34 | 0.31 | 8.29 | 1.45 | 2 | | | |
| | *G. hentscheli* | − | − | − | − | − | | | |
| | *S. rhaphidiophora* | − | − | − | − | − | | | |
| **Sediment** | **Layer** | | | | | | | | |
| KM | 0–16 cm | −21.42 | 1.46 | 5.87 | 0.77 | 6 | −527 (6000 a) | 2 | 1 |
| CM | 0–16 cm | −21.81 | 0.51 | 6.01 | 1.28 | 3 | | | |
| NM | 0–16 cm | −23.04 | 1.25 | 4.84 | 1.96 | 3 | −436 (4700 a) | 4 | 1 |
| CS | 0–16 cm | −21.57 | 0.58 | 5.65 | 1.13 | 3 | | | |
| **POM** | **Depth** | | | | | | | | |
| KM | 10 m | −27.43 | − | 5.55 | − | 1 | −36[a] | | |
| CM | 10 m | −29.51 | − | 4.17 | − | 1 | | | |
| NM | 10 m | −27.88 | − | 4.38 | − | 1 | | | |
| CS | 10 m | −26.98 | − | 4.99 | − | 1 | | | |
| **Zooplankton** | **Integration depth** | | | | | | | | |
| KM | 0–1258 m | −25.61 | 0.50 | 7.26 | 0.18 | 2 | | | |
| NM | 0–725 m | −25.74 | − | 7.41 | − | 1 | | | |
| KS slope | 0–1650 m | −26.23 | − | 7.62 | − | 1 | | | |
| **Fecal pellet** | **Integration depth** | | | | | | | | |
| KS slope | 50–100 m | −28.84 | − | 4.78 | − | 1 | | | |
| | 200–300 m | −29.16 | − | 3.31 | − | 1 | | | |
| | 300–400 m | −29.27 | − | 3.48 | − | 1 | | | |
| **Sea-ice matter** | **Depth** | | | | | | | | |
| KM | 0 | −29.32 | − | 0.98 | − | 1 | | | |
| **Associated macrofauna** | | | | | | | | | |
| KM | Bryozoans/Hydrozoans | −19.05 | 5.09 | 9.54 | 1.53 | 3 | | | |
| | Crustaceans | −21.67 | 1.71 | 12.35 | 0.89 | 2 | | | |
| | Asteroids | −13.40 | 6.05 | 10.84 | 1.53 | 2 | | | |
| | Siboglinid tubes | −19.77 | 0.29 | 8.91 | 1.02 | 3 | | | |
| | Serpulid tubes | −28.62 | 5.82 | 11.82 | 0.72 | 5 | | | |
| NM | Bryozoans/Hydrozoans | −12.12 | 3.34 | 10.37 | 0.60 | 1 | −245 (2300 a) | 2 | 1 |
| | Siboglinid tubes | −19.20 | 0.74 | 9.31 | 0.31 | 1 | −213 (1986 a) | 7 | 1 |
| | Serpulid tubes | − | − | − | − | − | −296 (1874 a) | 2 | 1 |
| | Bivalve shell | − | − | − | − | − | −587 (7162 a) | 1 | 1 |
| **DOM** | **Depth** | | | | | | | | |
| | 200–300 m | −22.3[b] | − | 4.6[c] | − | − | −406[b] | | |
| **DIM** | **Depth** | | | | | | | | |
| | 300 – 500 m | 0.85[a] | − | 4.7[d] | − | − | 34[b] | | |

$δ^{13}C$, $δ^{15}N$ and $Δ^{14}C$ values (‰) in sponges, potential food sources (suspended particulate organic matter POM, sea-ice matter, fecal pellet), zooplankton and associated macrofauna from the different summits (NM Northern mount, CM central mount, CS central mount saddle, KM Karasik seamount, and Kslope Karasik south slope). *n*. number of samples for each category. Values are provided as mean and standard deviation (sd) and for $Δ^{14}C$ the 1σ analytical error is shown. Please note that the $Δ^{14}C$ values of sediment belong to the first sediment layer (0–1 cm), while the stable isotope values were pooled together (0–16 cm). Radiocarbon years ago ($^{14}C$ a) for different sample types are also shown and reported for the organic $Δ^{14}C$ in *G. parva* (big specimen, KM), *G. parva* (small specimen, NM), siboglinid tubes and sediment samples, and the inorganic $Δ^{14}C$ in calcareous serpulid tubes, bryozoan skeleton, and bivalve shell. The analyzed crustaceans belong to the order of Amphipoda and Tanaidacea, and the fecal pellets were identified as *Oikopleura* sp. Dissolved organic (DOM) and inorganic (DIM including dissolved inorganic carbon and nitrate) matter values are from the literature.
[a]From ref. [93].
[b]From ref. [30].
[c]From ref. [104].
[d]From ref. [105].

representing 10–23%. Algae markers such as C20:4ω6/C20:5ω3 and C22:6ω3 represented less than 2% of the total PLFAs (Supplementary Table 3).

Among the bacteria-specific PLFAs, the $δ^{13}C$ values of the MBFA $C_{16}$ and $C_{18}$ were the highest (−19.5 ± 0.3‰), together with *i*C19:1ω12 (−18.8 ± 1.1‰), whereas the range of other bacteria markers including *i*C15:0 was between −22 to −27‰ (Fig. 5 and Supplementary Table 3). The $δ^{13}C$ of sponge-specific markers was more variable, varying between −17.9 and −25.7‰ (Fig. 5 and Supplementary Table 3).

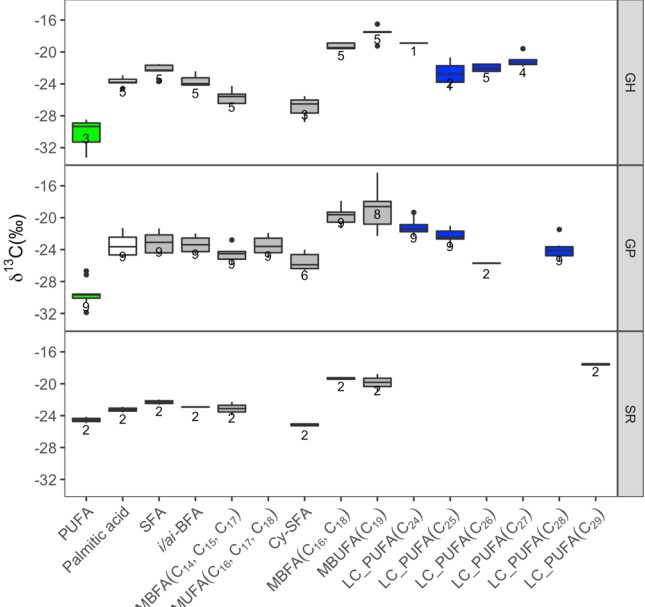

**Fig. 5 Compound-specific isotope analyses on sponge PLFAs.** Compound-specific δ[13]C (‰) on PLFA on the three sponge species analyzed: *G. parva* (GP) *G. hentscheli* (GH), and *S. rhaphidiophora* (SR). Different biomarkers were color code: palmitic acid (C16:0, white), bacteria-specific (gray), algae-specific (green), and sponge-specific (blue) markers. The lower and upper hinges of the boxes correspond to the 25th and 75th percentiles and the whiskers represent the 1.5x inter-quartile range (IQR) extending from the hinges. The median is indicated in the box plot as a solid line. The number in the plot indicates biological independent specimen. Source data are provided as a Source Data file. PUFA polyunsaturated fatty acids (FA), SFA straight-chain saturated FA, *i/ai*-BFA iso/anteiso branched C15-C17 FA, MBFA mid-methyl-branched FA, MUFA monounsaturated FA, Cy-SFA cyclopropyl-saturated FA, MBUFA mid-methyl-branched-unsaturated FA, LC_PUFA long-chain polyunsaturated FA.

**Sponge microbial community analyses.** According to amplicon analyses, the most dominant microbial phylum within *G. parva* was Chloroflexi ($51 \pm 14\%$ of the total microbial community), with SAR202 constituting $85 \pm 7\%$ of all Chloroflexi. Acidobacteriota and (α- and γ-) Proteobacteria were also abundant, constituting $17 \pm 5\%$ and $16 \pm 11\%$ of the total microbial comunity, respectively (Fig. 6a). Chloroflexi and Acidobacteriota were also significantly enriched in *G. parva* when compared with abundances in seawater reference samples (Fig. 6b). Among other significantly enriched microbial phyla were the known sponge symbiont clades Entotheonellaeota, Nitrospirota, and Nitrospinota. Additional metagenomic analysis revealed that *G. parva* hosted a substantial proportion of Poribacteria (5%) and the archaeal Nitrosopumilus (8%) (Fig. 6a).

A marker gene analysis of the *G. parva* metatranscriptome revealed a broad range of expressed pathways (Supplementary Data 1). For carbon metabolism, high expression levels were detected for refractory DOM utilization (Fig. 6c). Furthermore, carbohydrate-active enzymes were expressed. The function of the protein family (PFAM) representing methylamine degradation could not be unambiguously identified, but appears to be involved in GS-GOGAT pathways of ammonia assimilation rather than carbon metabolism. One chitinase (PF08329.10) was expressed in three of 13 sponge microbiomes, albeit at low levels (relative to average expression level $0.002 \pm 0.007\%$). Numerous proteases were expressed, 11 at the above-average expression level. A protease of the CLP family (PF00574.23) showed the highest expression levels and was related to an ATP-dependent protease

able to cleave a number of different proteins. With regard to nitrogen metabolism, distinct transcript signatures were detected for ammonia assimilation (via GS-GOGAT) and nitrate assimilation (assimilatory nitrate reduction to nitrite and nitrite reduction to ammonia). We found further high expression levels for several sulfur metabolism pathways, such as sulfatoacetaldehyde degradation and thiosulfate disproportionation (Fig. 6c). These metabolic properties are consistent with the organosulfur metabolism of pelagic Chloroflexi SAR202[30], but differ from typical sulfide-oxidizing symbionts known to occur in sponges of hydrothermal vents and seeps[31]. Accordingly, we did not detect any expression of typical bacterial genes for autotrophy in thiotrophs and for methanotrophs such as e.g., RuBisCO-encoding genes, *pmo*[32].

However, consistent with earlier studies of *Geodia* microbiomes, we found several other carbon fixation pathways in the metatranscriptome (Supplementary Data 2). These included the reductive citrate cycle, dicarboxylate-hydroxybutyrate cycle, hydroxypropionate-hydroxybutylate cycle, 3-hydroxypropionate bi-cycle, reductive acetyl-CoA pathway, and the phosphate acetyltransferase-acetate kinase pathway. These are represented in the mixotrophic genomes of Chloroflexi[33], Poribacteria[34], and Archaea[35], and suggest considerable potential for $CO_2$ fixation in the sponge symbionts consistent with the carbon isotope signature of the bacterial lipids[36].

## Discussion

Seamounts are often identified as "hotspots" of marine life due to enhanced productivity and particle export, with many supporting filter-feeding benthic communities, especially when associated with upwelling hydrodynamic conditions[37]. However, fluorometer and turbidity meter equipped Conductivity, Temperature, Depth (CTD) and ship-based Acoustic Doppler Current Profiler (ADCP) surveys conducted across the Langseth Ridge indicated very low productivity and sluggishness of currents ($<0.1$ m s$^{-1}$)[22,38], with no evidence for topography-enhanced primary production, retention of particles and/or substantially enhanced export fluxes indicated[39]. Any such enrichment would need very different hydrographic conditions than those observed[40].

As many seamounts result from volcanic or tectonic activity, some are known to degas and vent reduced compounds such as methane and sulfide[41]. Such active seamounts can support the establishment of chemosynthetic communities and locally enhanced microbial growth[42]. In the present study, we did not detect evidence of degassing or hydrothermal venting in the measurements obtained by our high-resolution CTD and turbidity surveys[22]. Additionally, the sponges sampled did not contain any of the known methane- and sulfur-oxidizing symbionts, nor did their bulk fatty acids show specifically depleted δ[13]C values[32]. However, we did observe and sample dense aggregations of tubes from an extinct chemosynthetic community of siboglinid tube worms. This blackened siboglinid tube detritus covered the peaks of the three surveyed summits. This material was reminiscent of active communities previously observed at Arctic cold seeps and vents, and recently also at sites in methane-rich Arctic shelf seas[26,43]. Given this observation, we tested the hypothesis that the sponges use mostly refractory detrital matter trapped in the spicule-tube mat, including material originating from these remnants of the now-extinct seep ecosystem.

**Carbon demand.** The sponge ground discovered at the Langseth Ridge is the most northerly and densest *Geodia* community reported to date[44]. It likely plays a crucial role in local biodiversity, benthic-pelagic coupling, and biogeochemical cycles.

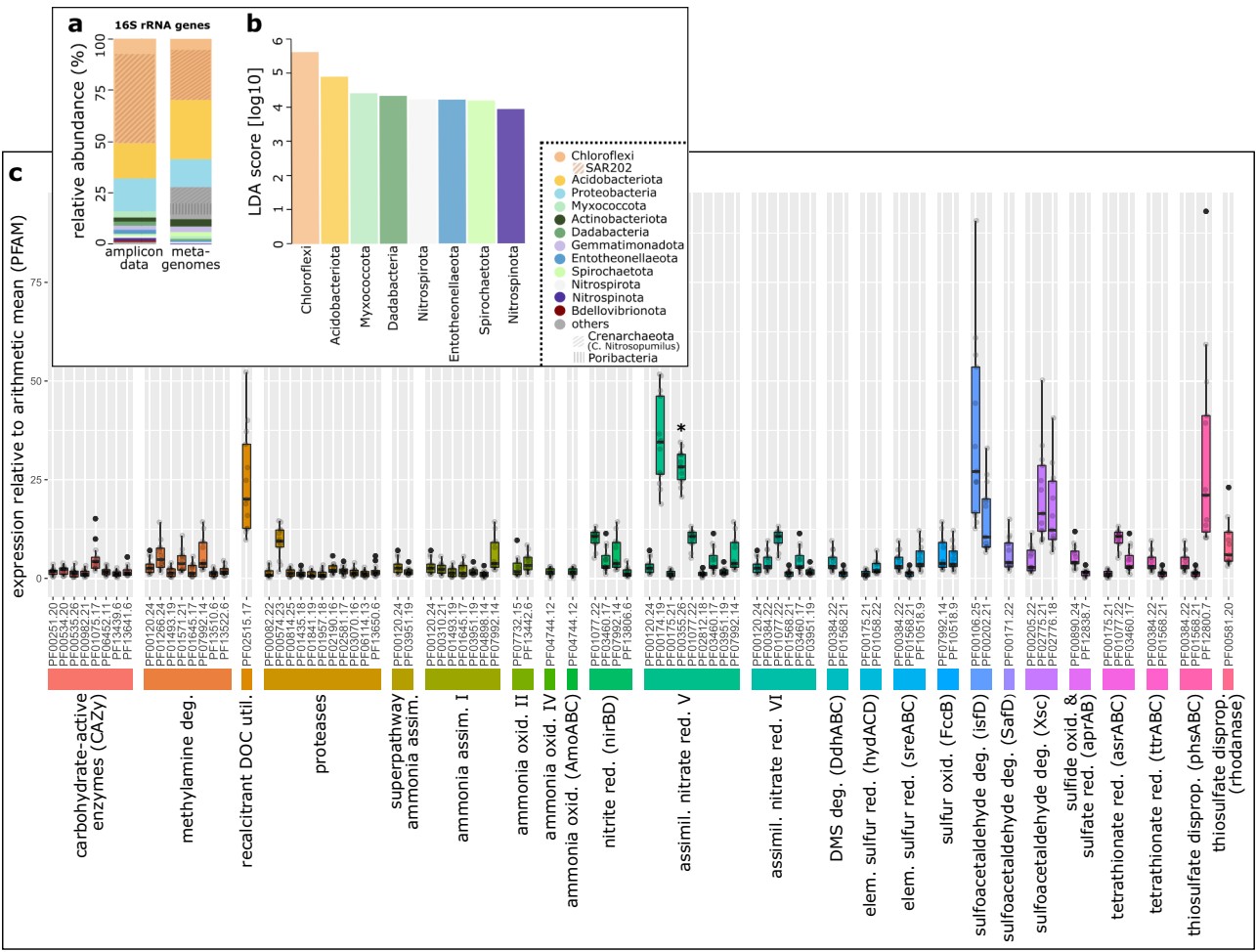

**Fig. 6 Microbial diversity and expression profiles. a** Average microbial community composition of *G. parva* (*n* = 11 biological replicates) based on microbial phylum level. Taxa in the legend are sorted in descending order of mean relative abundance in the amplicon data. The term "other" refers to microbial phyla with an average abundance below 1% in the amplicon dataset. In particular these are: *Bacteroidota, Deinococcota, Patescibacteria, Schekmanbacteria, Poribacteria, Verruocmicrobiota, PAUC34f, NB1-j, Calditrichota, Desulfobacterota, Margulisbacteria*, the SAR324 clade*, Firmicutes, Anck6*, and *MBNT15. Crenarchaeota* and *Poribacteria* are underrepresented in the amplicon data due to the amplification primers, but the high relative abundance of these phyla is shown in the metagenomic data. **b** Linear discriminant analysis (LDA) effect size plot based on amplicon data showing microbial phyla which are significantly enriched in *G. parva* in comparison to the ambient seawater. **c** Expression levels in the microbiomes of *G. parva* (*n* = 13 biological replicates). Shown are PFAMs with a mean expression above average and being either a marker gene in the MEBS analysis or representing a protease, chitinase, CAZy enzyme, or involved in refractory DOM utilization. Values for PF00355.26 (marked with an asterisk) were divided by 10 to fit a y-axis with higher resolution. The lower and upper hinges of the boxes correspond to the 25th and 75th percentiles and the whiskers represent the 1.5x inter-quartile range (IQR) extending from the hinges. The median is indicated in the box plot as solid line. assim. assimilation, assimil. assimilatory, deg. degradation, disprop. disproportionation, oxid. oxidation, red. reduction, util. utilization.

Most other Arctic and boreal systems exhibit substantially lower sponge densities than described here (<1 ind. m$^{-2}$)[8,11,45]. To date, Arctic benthic hotspots have been defined as areas with benthic biomass of >20 g C$_{org}$ m$^{-2}$ [46]. This is far higher than reported for most benthic communities within the Central Arctic, with those inhabiting depths of >500 m typically consisting of biomass <1 g C$_{org}$ m$^{-2}$ [21]. We estimated the sponge biomass alone to be 456 g C$_{org}$ m$^{-2}$ on average, with peaks of 1213 g C$_{org}$ m$^{-2}$ in the most densely populated areas (sponge coverage categories c and d across the flat seamount peaks). This biomass is on average 12-fold higher than reported for cold-water coral reefs on the Rockall Bank, where sponges represented 19% of living biomass (Northeast Atlantic mound system 55°29.7′N, 15°48′E, 38 g C$_{org}$ m$^{-2}$ sponge biomass)[47]. The biomass found at Langseth Ridge is comparable with biomasses measured at shallower sponge grounds of relatively high physical energy, such as a boreal *Geodia* ground at 300 m depth and exposed to bottom

current velocities of up to 27 cm s$^{-1}$ (6.3 ± 2.9 kg WW m$^{-2}$; 1.3 ± 0.6 kg DW m$^{-2}$; Norwegian Shelf 66°58′N, 11°06′W)[8], or on the Schulz Bank (up to 76 cm s$^{-1}$ [6]). Across the Langseth Ridge, bottom currents are substantially weaker (<few cm s$^{-1}$)[22,38]. The high density of juveniles observed in the proximity of sponge adults (3.4 ± 1.5 juveniles m$^{-2}$) shows that this *Geodia* community is actively propagating, suggesting that adult sponges have enough food and energy to allocate into their reproductive cycle. Demosponges are known to multiply by budding, which allows them to transfer directly the symbionts needed to utilize the relatively refractory matter available on the Langseth Ridge seamount summits[48].

Using mass-specific metabolic rates from the literature and taking into account the size of individuals[28], we estimate that this *Geodia* community pumps 1640 L water m$^{-2}$ per day across the most densely populated summit (KM), which means that in 1 year, 1 m$^2$ of this sponge community is able to process almost the

whole overlying water column up to 600 m depth. Assuming a respiratory quotient equal to 1[11,12], the estimated community metabolism may be assumed to be 25 mmol $O_2$ m$^{-2}$ day$^{-1}$ (averaged over the three summits), equivalent to a carbon demand of 110 g C m$^{-2}$ yr$^{-1}$ (ranging between 82 and 182 g C m$^{-2}$ yr$^{-1}$ in NM and KM, respectively). Our estimate is higher than that which has been estimated for Norwegian Sea sponge communities dominated by *Thenea* spp. (0.4–2.2 g C m$^{-2}$ yr$^{-1}$)[49], but within the range estimated for *Geodia* spp. communities (73 and 183 g C m$^{-2}$ yr$^{-1}$)[8,11] and some deep-sea cold-coral reefs (72–408 g C m$^{-2}$ yr$^{-1}$, RQ$_{(CO2:O2)}$ = 0.77)[8,47,50,51]. Even if we consider an RQ equal to 0.77, as recently estimated for deep-sea *G. barretti*[52], the average carbon demand to sustain the basal metabolism of this *Geodia* ground would then be approximately 85 g C m$^{-2}$ yr$^{-1}$, not considering the carbon demand of the other benthic life present. This is not matched by the particle export rate in this permanently sea-ice-covered region with low average primary productivity (from 1 to 25 g C m$^{-2}$ yr$^{-1}$)[19,20]. The particulate carbon flux reaching the seafloor in the area has been reported to be very low (<0.6 g C m$^{-2}$ yr$^{-1}$ at Nansen Basin)[53], and wholly insufficient to sustain the carbon demand of this sponge community. In addition, the carbon isotope values of the fresh particulate suspended matter, including sea-ice matter, suspended live (e.g., zooplankton) and detrital POM (e.g., fecal pellets) do not fit the values of the sponge community, (Fig. 4) and algae markers were observed to be relatively low in the sponge PLFA profiles (Supplementary Table 3). It should be noted that the sample size used for isotope analysis of the POM food sources was limited during the expedition PS101 for logistical reasons (*n* = 4, Table 1). However, episodic intense POM export events, as detected visually for the ice algae *Melosira arctica* in the central Arctic[54], were not observed at Langseth Ridge during the summer period. We cannot exclude the possibility that organic matter sinking from melting ice complements sponge nutrition[55], but the study area was outside the ice margin and for the past decades remained fully ice-covered year-round, putting forward the question of which additional food and energy sources are used to maintain such a high standing stock of life.

**Potential contribution by DOM and the role of the symbionts in sponge nutrition**. Recent investigations show that DOM assimilation may play the dominant role in sponge metabolism, both via pinocytosis of the host cells, especially the choanocytes, and in HMA sponges dominantly via the metabolism of their symbionts[56]. Deep-seawater masses are characterized by more refractory DOM[57], but also here, symbiont-mediated uptake of dissolved carbon and nitrogen has been found to play an important role in sponge metabolism[58]. This raises the question as to whether or not DOM could be a major source of carbon and nitrogen for the sponge grounds on the Langseth Ridge seamount peaks, where the ambient waters are rather poor in DOM (i.e., <50 μmol L$^{-1}$ in the intermediate and deep waters[59]; 70–80 μmol L$^{-1}$ in bottom waters[60]), and what DOM is present, being of a largely refractory quality.

The ability of *Geodia* to feed on DOM has been recently corroborated by an ex situ chamber experiment, showing that DOM ingestion can contribute over 90% of total organic carbon ingested[12]. Recent findings show that members of the Chloroflexi, and especially the SAR202 clade, can tap into dissolved refractory carbon sources[61]. However, the $\Delta^{14}$C signature of background DOM in the Arctic of around −400‰ at intermediate water depths excludes the possibility of a substantial contribution of this DOM to sponge and symbiont biomass given their relatively high $\Delta^{14}$C value. Even though metabolic estimates on *G. barretti* indicated that 76% of the ingested DOC is respired[52], the

relatively high sponge $\Delta^{14}$C value suggests a mechanism by which a substantial fraction of relatively "young" organic carbon is incorporated by the holobiont, likely through DIC assimilation (Supplementary Data 2).

The sponge microbiome of *G. parva* species investigated here was mostly composed of Chloroflexi, with class SAR202 accounting for 80% of this phylum and >40% of the whole microbial community, higher percentages than found in previous studies of deep-water *Geodia*[14,62], but similar to *Craniella* grounds from the boreal region[63]. Poribacteria were also present as indicated by their representation in the metagenome (Fig. 5a). It has recently been suggested that Chloroflexi and especially SAR202 degrade complex hydrocarbon compounds by hydrolysis with extracellular enzymes[61], and are equipped to degrade chitin[64]. Many of the enzymes involved in the degradation of such complex compounds have indeed been found in SAR202 symbionts and Poribacteria of the Mediterranean *Aplysina aerophoba*[33,34]. The metabolic profile of expressed genes supports a symbiont metabolism that is based on refractory DOM as a carbon source. Interestingly, based on the many sulfur–related genes expressed, this may include sulfur-enriched proteins and sulfurized OM. This would fit the hypothesis of substantial carbon and energy intake of the local accumulations of blackened, sulfurized siboglinid tube matter on which the sponges across the Langseth Ridge are most commonly observed.

In addition to heterotrophic processes including the use of sulfurized matter, such as in DMS degradation and reduction reactions, we found that *Geodia* symbionts have a noticeable expression of genes potentially related to sulfur cycling and autotrophy. This is consistent with previous reports of sulfate reduction and sulfide oxidation in incubation experiments of *G. barretti*, which can host sulfate-reducing Deltaproteobacteria, some of which grow autotrophically[16]. The expressed gene profile of nitrogen metabolism also shows an increased gene activity for heterotrophic nitrate, ammonia assimilation, and oxidation. Previously, almost the entire microbial nitrogen cycle has been annotated in the *G. barretti* microbial community, including aerobic nitrification as well as anaerobic denitrification and ammonium oxidation[16,65]. In the only other transcriptome study on deep-sea sponges realized to date, similarly, high expression levels of ammonia assimilation-related genes were observed in the *G. barretti* microbiome[66]. The relatively lower N isotope signal within the studied Demosponges when compared to the symbiont-free deep-sea glass sponges (ca. 8‰ and 16‰ $\delta^{15}$N, respectively)[67] is a further indication that N is recycled within sponge tissue by its symbionts.

Finally, the comparison of the $\Delta^{14}$C biomass value of the sponge and its bacteria (assessed by the representative bacterial FA *i*C15:0) suggests a substantial source of post-bomb DIC (i.e., $\Delta^{14}$C > 0‰), pointing to autotrophic growth. Accordingly, the metatranscriptome indicates a substantial potential for $CO_2$ fixation by the presence of several complete autotrophic pathways (Supplementary Data 2). Previous studies of sponge symbionts Poribacteria, Chloroflexi, and SAR202 genomes suggest that these may be important enablers of deep-water biogeochemical processes linking C, S, and N metabolism, including autotrophy[30,33]. The predominance of MBFAs suggests production by distinct sponge-specific bacteria such as Poribacteria[68], as has been found in other boreal deep-sea sponges[68,69]. The bacterial PLFAs $\delta^{13}$C values are not indicative of substantial methylotrophy or thiotrophy, but rather other forms of autotrophy in MBFAs[69]. The isotope values (Fig. 3) and results of a detailed study of the lipidome of these species[69] suggest that these bacteriosponges complement their nutrition by digesting their symbionts for essential nutrients and organic precursors of their biomass.

**The role of fossil detrital matter in the nutrition of the sponge community**. The $\delta^{13}$C ($-18.4 \pm 1.4$‰), $\delta^{15}$N ($8.1 \pm 0.7$‰) and $\Delta^{14}$C ($-16 \pm 6$‰) values of the Demosponges investigated here point to a complex food web interaction and mixed detrital food sources. We investigated this further using an isotope mixing model (SIAR)[70], using different end-member values to account for the different potential food sources, with two stable isotopes. To reflect that the sponges rely on the capacity of their microbiomes to digest refractory POM, including detrital and sulfurized chitin and protein, we chose the following trophic transfer fractionation factors $0.5 \pm 0.5$ ‰ for $^{13}$C and $1.5 \pm 0.5$% for $^{15}$N, considering a substantial microbial contribution to the holobiont metabolism[67,71], with the result of a rather high proportion of siboglinid tube material (ca. 40%). When using the classical fractionation factor for predatory animals of $0.5 \pm 0.5$ ‰ for $^{13}$C and $3.5 \pm 0.5$% for $^{15}$N[72], this proportion would decrease substantially, and more suspended POM would be needed. Given the uncertainty of potentially high internal N recycling by the sponge holobionts, the model cannot fully constrain the composition of food sources, which may also vary during the lifetime of individual sponges. With this in mind, the differences between the isotopic signatures of sponge juveniles and adults could be explained by different modes of nutrition. In particular, the lower $\delta^{15}$N found in juveniles when compared to the adults (Supplementary Fig. 4b) may suggest less nitrogen recycling in young sponge holobionts, pointing to the importance of considering the age of a sponge in the determination of sources of bulk isotopes in a long-lived organism such as sponges.

Interestingly, the closest carbon and nitrogen stable isotope values to the sampled sponge tissue measured in the current study were those of the siboglinid tubes. The tubes of these gutless annelids consisted primarily of chitin and refractory proteinaceous matter. Studies of siboglinids from an active Barents Sea methane seep at >300 m depth show that their tubes can reach a $\delta^{13}$C of $-38$‰ and $\delta^{15}$N of $-1.7$‰, given background POM and sediment $\delta^{13}$C values of $-25$‰ and $3.9$‰, respectively[73]. Previous assessments of the production, composition, and degradation of siboglinid worm tubes indicated that active communities could produce hundreds of grams of chitin and proteinaceous matter per $m^{-2}$ $yr^{-1}$ and that their chitin-protein tube material is relatively refractory due to sulfurization when compared to other chitin sources, such as crustacean carapaces[74]. The sponges were found arrayed across the blackened tube detrital matter, which together with the abundant sponge spicules formed the dense underlying mat observed across the seamount peaks. Empty tubes of the genus *Polybrachia* represented the most abundant detrital matter within this matrix, composed of sulfurized proteinaceous and chitin-rich polymers. Assuming a spicule-tube mat average thickness of >4 cm, the amount of OM (e.g., the mixture of siboglinid tubes and deposited POM) trapped within this matrix may be estimated to be ca. 500 g C $m^{-2}$. Hence, we propose that the sponges are opportunistically using this nitrogen-enriched fossil detritus as a food source, the microbes in sediment (and possibly sponge-associated symbionts[64]) solubilize chitin, proteins, and carbohydrates, and the dissolved organic matter released is then used by the sponge holobionts (Fig. S5).

Chitin and proteinaceous matter are important sources of C and N for marine organisms, with bacteria commonly the main agents of chitin degradation in many aquatic ecosystems[75]. In deep-sea invertebrate guts, chitinolytic bacteria play an important role in chitin digestion[76], and chitinolytic activity was observed in associated bacteria in carnivorous[77], as well as Demospongiae species[64]. It is not known if sponges themselves can produce chitinases for digestion, but some species may be able to, as they can produce chitin[78]. The digestion of detrital matter in the mat, and especially the empty siboglinid tube material, may be aided by the sponge-associated microbes shown to express chitinases and proteases as well as a suite of enzymes related to sulfur- and refractory carbon metabolism. It is highly likely that the microbial processes in the mat and sponges are essential in recycling the nitrogen and sulfur-enriched polymeric matter and in producing dissolved compounds that can be taken up by the holobiont.

The radiocarbon value of the sponges clearly indicates a mixed carbon source, including a substantial fraction of DIC, as both the fossil matter and DOM are too low in $\Delta^{14}$C to account for all the carbon. The $\Delta^{14}$C values measured from the sponge tissues represent the mean sample age and therefore our findings do not unequivocally show assimilation of an ancient or recent C source as the bulk $\Delta^{14}$C, but an amalgamation of C of different ages (Fig. S5).

**Sponges associated with fossil seep detritus**. Our $\Delta^{14}$C dating results indicated that the *Polybrachia* sp. community was present and active 2000–3000 years before the sponges colonized the area. Accordingly, the radiocarbon age of sedimentary OM from beneath the spicule-tube mat was considerably older than that measured from within sediments directly exposed to the water column (Table 1), suggesting that the sponges settled on and colonized successfully the extinct seep community remains. Colonization of low or inactive hydrothermal sites by pioneer non-chemosynthetic fauna may occur over >1000 a following the cessation of venting[79]. The shells of bivalves and some tubes may remain in situ for thousands of years if favorable environmental conditions are met (e.g., low sedimentation rates, low-temperature conditions, and depths shallower than those associated with the local carbonate compensation depth)[80]; environmental constraints which are fulfilled across the Langseth Ridge summits.

It is noteworthy that the largest sponges were observed only within the central regions of the seamount peaks, where deposits of siboglinid tubes were observed and were always associated with substantial spicule abundances. The spicule-tube mat extended beyond this seamount central region of high sponge coverage, though as a thinner layer with far less sponge spicule inclusion. Across this exposed substrate individual sponges seemed to actively crawl on top of the spicule-tube mat, leaving behind abundant and visible spicule trails[81]. Although sponges are generally considered sessile organisms in adult life, locomotion has been observed in few species and attributed to be a response to food availability and/or population density[82]. We suggest that this mobility behavior facilitates foraging, whilst simultaneously promoting localized detrital accumulation for later utilization by the sponges. With movement, these sponges deposit large numbers of spicules which form the dense mat-like structure so prevalent on the Langseth Ridge peaks. These spicules are locally integrated with the fossil detritus from the extinct hydrothermal ecosystem, increasing seafloor rugosity and roughness and therefore promoting the local settling of particles and biogenic materials, then subsequently hydrodynamically entrapping this material in the complex mat matrix. This mat can build up to decimeter thickness over time. The OM trapped in this complex mat matrix, including sediment-borne bacteria, may at times be remixed through sediment resuspension by bottom water currents. Such deposit mixing was previously observed along continental shelf edges[67,83,84]. However, in the studied open ocean seamount area, the sluggishness of the bottom current may make such remixing a rare occasion[22,38]. The observed repositioning of the sponges may potentially facilitate their access to recently integrated detrital matter with this movement also potentially increasing the flux of small particles and microbially

hydrolyzed DOM to the sponges[33,34], which can then be used by the holobionts.

The dense sponge grounds discovered here represent an astonishingly rich ecosystem demonstrating the ability of sponges and associated microorganisms, here foremost represented by members of the Chloroflexi, to exploit a variety of refractory food sources including fossil seep detritus. We hypothesize that the remnants of an extinct degassing seamount chemosynthetic community may have facilitated the settling and successful colonization by sponges across the northerly Langseth Ridge seamount structure of the Gakkel Ridge, by providing a suitable substratum for settlement, detrital entrapment, as well as being a food source per se. Today, a number of vents and seep systems are known from the wider Arctic region, e.g., in the Barents Sea, where these host reef communities are composed of siboglinids, sponges, and deep-water corals[25,26]. Prior to our study, no actively venting seep community or similar sponge ground had been identified in the high Central Arctic >80°N, an area of the ice-covered ocean which remains understudied given the difficulties associated with observing and sampling such ice-covered deep-sea ecosystems. With sea-ice cover rapidly declining and the ocean environment changing, a better knowledge of hotspot ecosystems is essential for protecting and managing the unique diversity of these Arctic seas.

## Methods

**Study site**. The Langseth Ridge forms a north-south aligned V-shaped ridge structure on both sides of the Gakkel Ridge, extending from 87°N 62°E to 85°55′N 57°45′E for ~125 km in the central Arctic Ocean[22]. The major ridge axis is oriented northwest-southeast, rising from >2500 m depth at its southeasterly tip, to the shallower depths of 500–700 m at the summits. The northern end next to the axial valley of the Gakkel Ridge is very steep, descending to >5000 m water depth. The Langseth Ridge consists of three summits: Northern Mount (NM, 86°51.86′N 61°34′E, mean summit depth 630 m), Central Mount (CM, 86°47.83′N 61°54.52′E at 722 m) and Karasik Seamount (KM, 86°42.38′N 61°08.07′E, summit at 585 m). The KM and CM are separated by a saddle structure referred to hereafter as the Central Mount saddle (CS, 1055 m) (Fig. 1a).

The seamount slopes are dominated by bare bedrock outcrops, whereas the summits are to a large extent covered by dense sponge aggregations (Figs. 1b, 2). The seafloor is barely visible between dense sponge aggregations and a mix of underlying sponge spicules and dead worm tube detrital matter. Where visible, the seafloor comprises sediment-dusted basalts. From the flat seamount peaks downslope, the sponge community extends to ~1000 m water depth, with aggregations mostly observed on steeper flank surfaces of small, near-horizontal rocky outcrops. Geological sampling indicated that the seamounts consist primarily of basalt rock with brecciated rock inclusions of volcanic rock clasts[22]. Rock samples from the northern slope showed hydrothermally altered materials with disseminated sulfide precipitates cut by sulfate or carbonate veins[22]. The current velocities measured above the ridge during the cruise were low (<0.1 m s$^{-1}$) with a predominantly westward component in flow direction and no evidence of upwelling currents detected. Mean-bottom water temperature was close to zero degrees Celsius and salinity was around 35 at the time of survey[38].

**Sample collection**. Samples were obtained by RV *Polarstern* (AWI Expedition PS101) during September-October 2016. Sponge samples, together with spicule-tube mat, sediment, and associated macrofauna were collected via camera-guided multiple corer (TV-MUC), USNEL box-corer (0.25 m$^2$), chain bag dredge and "Nereid Under Ice" remotely operated vehicle (NUI ROV), and seawater samples were obtained from CTD-Rosette casts. Sampling gear was equipped with POSI-DONIA transponders for geo-referencing. Sponges, associated macrofauna, spicule-tube mat, and sediment samples for isotope analyses were immediately stored on retrieval to surface at −20 °C. Subsamples of sponge tissue for amplicon sequencing, metagenomic, and metatranscriptomics were frozen at −80 °C and fixed in RNA later, respectively. As controls, seawater samples were collected in 4 technical replicates: 2 L of seawater were filtered through 0.2 μm Durapore PVDF 47 mm membrane filters (seawater filter – SWF), which were frozen at −80 °C prior to DNA extraction. Sponge subsamples were fixed in ethanol for taxonomic identification via sponge spicule microscopy and barcoding. Small macrofauna, bivalve shells, and polychaete tubes were sorted and identified to the lowest possible taxonomic level.

Zooplankton and fecal pellets of *Oikopleura* sp. were collected using a side net (60 cm, 200–300 μm mesh size) attached to the outside of a larger multinet from water column sampling (0–1650 m). Particulate organic matter (POM) was collected from 10 m depth using CTD-rosette and filtered through Whatman GF/F filters (0.7 μm, 10–14 liters per filter). All filters were stored at −20 °C.

A detailed description of sample collection and the cruise expedition overall can be found in Boetius and Purser (2017)[22]. Sample categories and sites of the collection are shown in Fig. 1a (for a detailed list of all stations, sample types and analyses see Supplementary Data 3).

**Sponge abundance, biomass estimations, and seafloor characterization**. The sponge abundance was visually characterized through image analysis. Four Ocean Floor Observation and Bathymetry System (OFOBS) deployments (cruise stations 89, 100, 120, 169) collected 3.2 km of video, still image and sidescan data transects. All OFOBS images collected during the *Polarstern* PS101 cruise are available at: https://doi.org/10.1594/PANGAEA.871550. The OFOBS camera sled was operated at a flight height of 1.5 m above the seafloor, as described in ref. [85]. A total of 696 images were analyzed throughout these transects. All images from the abundant sponge ground imaged during the station 169 deployment were analyzed, whereas every tenth image was analyzed across the other ridge stations. The mean seafloor area recorded in each image was 18.5 m$^2$ (median 4.5 m$^2$), translating to a total of 13,000 m$^2$ of the surveyed area. The sizable difference between the mean and median image coverage averages is the result of the extreme topography imaged in the category d areas of the Langseth Ridge; the steeply sloped flanks. Across habitat categories a, b, and c, a much more uniform seafloor coverage of ~4 m$^2$ per image was maintained as the seafloor slope angle was minimal. In each analyzed image, all visible sponge individuals were counted and measured (length and width) using the PAPARAZZI software application[85,86] for later volume and weight determination, except for very small sponges (<1 cm of diameter), which were simply counted and quantified as juveniles. Sponge biomass was estimated based on size-volume-weight conversion factor from the literature (biomass $L = 0.0003 \times$ length$^3$)[11,28].

It was not possible to distinguish between the two most abundant genera (*Geodia* and *Stelletta*) on images due to their similar morphology; therefore, they were annotated as Demosponges in the analysis. Glass sponges were also quantified when present. In addition, associated macrofauna such as asteroids, cnidarians, crustaceans, fishes, and unknown fauna were also logged.

Throughout each OFOBS deployment accurate positioning of the platform was made by utilizing the POSIDONIA Ultra Short Base Line (USBL) system used by *Polarstern*, augmented with inertial navigation data from the OFOBS integrated PHINS inertial navigation and Dynamic Velocity Logger (DVL) system. The depth of the seafloor recorded in each image could be accurately determined to within a few cm, with a lateral position accuracy of a few tens of meters[85]. From this positioning data, the aspect, rugosity, and slope terrain variables associated with each image were computed by comparing measurements of the depth and position stamps recorded at the time of each subsequent image collection.

In addition to logging all fauna within each image, and sizing any sponges present, each image was assigned as best representing one of four sponge distribution categories: (a) occasional sponges present on shelves abutting steep walls; (b) siboglinid tube mats exposed, with occasional sponges; (c) spicule mats abundant, with numerous sponges also present; (d) very dense sponge community on spicule-mat—most of the image area covered by live sponges. Category c and d were mostly observed across the flat summits of the Langseth Ridge. The area covered by the flanks of the Langseth Ridge seamount could not be readily determined from the collected data. The areas of the Langseth Ridge with denser sponge coverage, i.e., categories b, c, and d were: category b, ca. 10 km$^2$, category c, ca. 15 km$^2$, and category d, ca. 2.5 km$^2$.

The ash-free dry weight (AFDW), as a proxy of the organic matter (OM) content of the spicule-tube-mat layer, was estimated by subtracting the ash weight (after 12 h combustion of the dry weight samples in a muffle furnace at 500 °C) from the dry weight (DW, 100 °C, 24 h) of a known volume. To estimate the extent of the spicule-tube-mat layer, sediment cores were photographed adjacent to a ruler (Supplementary Fig. 1) and subsequently measured using manual delineation with Image J (Ver. 64). Since volume-specific metabolism may largely vary between sponges of different sizes[87], the volume of water processed and oxygen respired per day by this sponge community was calculated as a function of sponge volume using allometric relationships estimated for *G. barretti* from the literature ($L_{pumped} = 0.4952 \times V_{sponge}^{-0.52}$ [28] and $O_2 = 25.5 \times V_{sponge}^{-0.23}$ [52]). No temperature correction was applied, since no effect of temperature (from 0 to 15 °C) was observed in the respiration rate of cold-water massive sponges[12]. It is important to consider that the minimal respiratory carbon demand was estimated only for the sponges, thus our estimates are conservative and the total carbon demand for the ecosystem as a whole may be significantly higher.

**Bulk stable isotope (SI) and compound-specific isotope analyses (CSIA)**. In order to investigate the food sources used by sponges, stable C and N bulk isotopes were analyzed from sponges, as well as from putative food sources such as suspended POM from overlying waters, sediment, fecal pellets and sea-ice material. The bulk SI analysis was also performed on the remnant dead biomass (siboglinid tubes) and associated macrofauna (hydrozoans, bryozoans, crustaceans, asteroids, and serpulid tubes) in order to characterize the entire community. Extreme care was taken to avoid contamination of sponge samples from siboglinid tubes as epibionts were removed, and only internal sponge tissue (i.e., no cortex) was used in our analysis. Sample preparation methodologies for SI analysis was

sample-type-dependent. A detailed description of the methods and statistical analyses can be found in the supplementary Material and Methods.

Phospholipid-derived fatty acids (PLFA) were prepared for isotope analysis in three main steps: (i) modified Bligh-Dyer-Extraction; (ii) chromatographic separation, and (iii) derivatization of functionalized compounds. Details of the step-by-step protocol for PLFAs extraction and identification are delineated in our companion paper[69] with the associated protocol available online (https://doi.org/10.17504/protocols.io.bhnpj5dn).

In this study, we used PLFA as organism-specific markers for the identification of food sources. Based on the literature the following PLFAs were considered: iso/anteiso branched $C_{15}$-$C_{17}$ ($i/ai$-BFA), monounsaturated fatty acids (MUFA $C_{16}$-$C_{18}$), mid-methyl-branched fatty acids (MBFA $C_{14}$-$C_{18}$), mid-methyl-branched-unsaturated fatty acids (MBUFA $C_{19}$), and cyclopropyl-saturated fatty acids (cy-SFA) as bacteria-specific[88]; polyunsaturated fatty acids (PUFA, 20:4ω6, 20:5ω3, and 22:6ω3) as algae-specific[89]; and long-chain polyunsaturated fatty acids (LC-PUFA $C_{24}$-$C_{29}$) as sponge-specific[69,90]. Straight-chain saturated fatty acids (SFA) were considered as general markers.

**Radiocarbon dating $\Delta^{14}C$.** The $\Delta^{14}C$ value of organic carbon was measured in *G. parva* tissue, siboglinid tubes, sediment samples (0–1 cm layer), and isolated PLFAs in order to trace putative carbon food sources for sponges. In addition, the $\Delta^{14}C$ value of inorganic carbon was measured from the calcium carbonate of serpulids tubes, bryozoans, and shells of the bivalve *Limatula hyperborea*. A detailed description of the standard methodology is provided in the supplementary Material and Methods and in ref. [91]. Regarding sponge tissue, one large and one small (<5 cm diameter) individual were selected for analysis, to facilitate a relative age comparison between the two specimens, assuming that the small individual was younger than the larger sponge. Since high $\Delta^{14}C$ variability can be observed within the same sponge individual along the axis of growth[92], we assumed that *G. parva* growth to be radial, and different subsamples were selected to account for this by collecting successive sections from below the cortex (the cortex was excluded) towards the center in the larger specimen, whereas from the smaller sponge two pieces were randomly selected. In order to infer putative deposition rates and examine whether the spicule-mat layer may act as OM trapping, isolating the sediment below from access, we selected two contrasting sediment samples. The first sample was collected from below an expanse of the spicule-mat like structure found across the flat central regions of each peak, which mainly consisted of chitin tube detrital matter intermixed with spicules, with the second sample taken from a spicule-mat free area (i.e., the uppermost layer of sediment was exposed to the water column). The $\Delta^{14}C$ of the DOM, DIM (dissolved inorganic matter including DIC + nitrate), and suspended POM were taken from previous studies of nearby sites in the Central Arctic (88°8.96′N, 78°14.56′E, and 77°59.859′N, 150°4.887′W) and are −406[29], 34[29] and −36‰[93], respectively.

**Amplicon, metagenomics, and metatranscriptomics analyses.** *G. parva*, the most abundant sponge species inhabiting the Langseth Ridge, was selected for amplicon, metagenomic and metatranscriptomic analyses. A detailed description of the amplicon workflow is included in the supplementary Material and Methods. Briefly, the V3V4 variable regions of the 16 S rRNA gene were amplified with the primer pair 341F-806R and sequenced on a MiSeq platform (Illumina). Exact Amplicon Sequence Variants (ASVs) were generated with the help of the DADA2 algorithm[94] and classified based on the Silva database. The Linear Discriminant Analysis (LDA) Effect Size (LEfSe) algorithm[95] was applied to determine which microbial phyla were significantly enriched in *G. parva*, when compared with those measured in the ambient seawater, and to rank those phyla according to estimated effect sizes.

Metagenomic DNA was co-extracted alongside RNA from 13 sponge samples (Supplementary Data 3) in technical replicates with the AllPrep DNA/RNA Mini Kit (Qiagen, Cat. No. 80004) including a DNase treatment step using the RNase-free DNase Set (Qiagen, Cat. No. 79254). After extraction, RNase was inhibited with SUPERase In (Ambion, Cat. No. AM2694). Based on NanoDrop values, the best two of four technical replicates were pooled, cleaned, and concentrated for each sample with the Qiagen RNeasy MinElute Cleanup Kit (Qiagen, Cat. No. 74204). A second DNase treatment was applied (DNA-free Kit for DNase Treatment and Removal, Ambion, Cat. No. AM1906) to remove the remaining DNA. Eukaryotic mRNA was removed with Dynabeads mRNA DIRECT Purification Kit (Invitrogen, Cat. No. 61012) (keeping the supernatant for further steps, as it contains the microbial mRNA), followed by a second cleanup with the Qiagen MinElute Kit. Finally, all rRNA (bacterial and eukaryotic) was removed with the Ribo-Zero Gold rRNA Removal Epidemiology Kit (Illumina, Cat. No. MRZE724) using the Qiagen MinElute Cleanup Kit for the final cleanup as described in the Ribo-Zero protocol. RNA quality and quantity were assessed by NanoDrop, Experion, and Qubit assays. Transcription, sequencing library construction, and Illumina HiSeq sequencing were conducted at the Institute for Clinical Molecular Biology (IKMB) of Kiel University. Sequence quality of all read files was assessed with FastQC. Metagenomic reads were trimmed with Trimmomatic v0.36 (ILLUMINACLIP:NexteraPE-PE.fa:2:30:10 LEADING:3 TRAILING:3 SLIDINGWINDOW:4:15 MINLEN:36)[96] and co-assembled with megahit v1.2.9[97]. Small subunit (SSU) ribosomal genes were identified with the ssu_finder function of CheckM v1.0.12[98] and taxonomically annotated with the SINA. The raw metatranscriptomic reads were trimmed with Trimmomatic v0.36

(ILLUMINACLIP:Truseq3-PE2.fa:2:30:10 LEADING:3 TRAILING:3 SLIDINGWINDOW:4:15 MINLEN:36)[96] and co-assembled de novo with Trinity v2.0.6 in paired mode[99]. Coding regions were identified with TransDecoder v5.5.0 (https://github.com/TransDecoder/TransDecoder/releases) and PFAMs were annotated with Trinotate v3.2.0[100,101].

Within Trinity, expression levels were quantified using bowtie2[102], RNA-Seq by Expectation-Maximization (RSEM, http://deweylab.github.io/RSEM/) including transcripts per million transcripts (TPM) normalization within each sample, and cross-sample normalized by the trimmed mean of log expression ratios (TMM). Within each sample, the resulting expression values were summed for each PFAM and divided by the overall PFAM-level arithmetic mean of 7.500434. MEBS v1.2[103] was used for completeness estimations of various metabolic pathways. Different PFAMs databases were assessed for enzymes involved in heterotrophy, including hydrolytic enzymes such as chitinases, proteinases, carbohydrate-active enzymes (CAZy), and acyl-CoA transferase CaiB and other family III transferases (PF02515), flavin-dependent oxidoreductase of the luciferase family (PF00296.20), and short-chain alcohol dehydrogenase (PF00106.25), which have previously been identified as key enzymes in microbial utilization of refractory DOM specifically by SAR202[33,61].

**Reporting Summary**. Further information on research design is available in the Nature Research Reporting Summary linked to this article.

## Data availability

Source data are provided with this paper. The isotope data generated in this study are provided in the Supplementary Information and Source Data file. Seabed images used to estimate the sponge community biomass are publicly available from the PANGEA data archive (https://doi.org/10.1594/PANGAEA.871550). The processed density and biomass data are available in the OSF.IO database under the project Deep-Sea Arctic Geodia ground: sponge count and biomass (https://doi.org/10.17605/OSF.IO/VCXYE). Silva data used to classify amplicon sequences is available at https://www.arb-silva.de/. Metagenomic, metatranscriptomic, and amplicon data were deposited in the NCBI database under BioProject PRJNA454581 with accession code SRR7182305-17 for metatranscriptomic raw data and GIMA00000000 for metatranscriptomic co-assembly. The original cruise report is available at https://epic.awi.de/id/eprint/44286/ with the description and links to related published data from the field study.

## Code availability

All software utilized in this analysis are available online and are referenced in here or supplementary information. The R scripts for isotope and PLFA analyses have been deposited in the Open Science Foundation database, and are publicly available at https://osf.io/hxtjn/

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

## Acknowledgements
We thank the captain and crew of PS101 and the OFOBS and NUI teams for their excellent support at sea. We thank the GeoLab staff from the Utrecht University, in particular the research assistant Arnold van Dijk, Dr. Klaas Nierop, and Desmond Eefting for their support in the isotope laboratory. We thank Elizabeth Bonk from AWI for her technical support for the radiocarbon measurements. We thank Jan Steger from the University of Vienna for the identification of bivalve shells. This paper is dedicated to the memory of Hans Tore Rapp. This study received funding from the DFG Cluster of Excellence "The Ocean in the Earth System at the University of Bremen, grant. 49926684, from the ERC Adv Grant ABYSS to AB (grant no. 294757) and from the European Union's Horizon 2020 research and innovation program under grant agreement No. 679849 (SponGES). Additional funds came from the Helmholtz Association, Max Planck Society, and the Netherlands Earth System Science Center.

## Author contributions
A.B. conceived the project; A.B. and A.P. carried out the surveys; B.M.S., J.D., and A.P. collected the samples; T.M.M., B.M.S., K.B., A.D.K., A.P., H.G., and G.M. processed the samples and analyzed the data; H.T.R. and J.D. did the taxonomic identification; T.M.M. and A.B. were the primary interpreters of the data and writers of the manuscript with contributions from all co-authors, in particular from J.M.M. and U.H.

## Funding

## Competing interests
The authors declare no competing interests.

## Additional information

**Peer review information** *Nature Communications* thanks Bodil Bluhm, Maxim Rubin-Blum and the other anonymous reviewer(s) for their contribution to the peer review this work. Peer reviewer reports are available.

