## [Peer Review File · Nature Communications]

Reviewers' Comments:

Reviewer #1:

Remarks to the Author:

This beautiful study investigates the food sources, and the mechanisms of their assimilation by sponges from a fascinating location – the deep-sea Arctic seamounts. The fact that the authors collected samples from the seabed below the deep ice-covered waters is outstanding, given the technological effort behind the findings. This study uses a comprehensive bulk and compound-specific isotopic analysis, to show the contribution of DOM and DIC to the diet of sponge holobiont, and associates sponge diet with fossil seep detritus. Using omics, the authors highlight the contribution of microbial partners to sponge holobiont nutrition. Overall, this is a wonderful scientific effort, describing the functionality of hotspots of life in a rapidly-changing arctic environment.

My main concern is that the authors do not refer to the metagenomes when conducting the functional analyses of the community. In general, the information regarding the metagenomic sequencing appears to be lacking (how deeply were the metagenomes sequenced?). Did the authors try to bin the metagenome-assembled genomes, especially those of SAR202 and look at the expression profiles of specific microbes? Although the metatranscriptomes provide ample information regarding the microbial community function, metagenomics could also contribute to the understanding of the metabolic potential.

Minor comments:

L186: Nitrosopumilus archaea – Candidatus prefix doesn't seem to be necessary, as Nitrosopumilus species have been cultivated and classified (https://www.microbiologyresearch.org/docserver/fulltext/ijsem/69/7/1892_ijsem003360.pdf?expires=1624340739&id=id&accname=guest&checksum=7204B3609C98A310994C2C51A9B5CCF5)

L203: Please consider the enzymes of the HP/HB cycle in archaeal Nitrosopumilus autotrophs.

L204: RuBisCO is not a gene. Please italicize *cbbLM*, *pmo*. How about *cbbM*?

L207: There is no need to capitalize the cycle names.

L225: ...methane- and sulfur-oxidizing symbionts. MOX and SOX are abbreviations not known to the readership (better acronyms are MOB and SOB, for methane- and sulfur-oxidizing bacteria).

L231: Please rephrase "on which they sit".

L321: "...including autotrophy"?

L339: Please remove a dot.

Maxim Rubin-Blum

Reviewer #2:

Remarks to the Author:

The authors present very spectacular findings of dense sponge aggregations on seamounts in the central Arctic Ocean where food supply from water column production and vertical flux is extremely low and not sufficient to support those sponge densities. Those of us who followed the media of this expedition have been waiting for manuscripts like this one for some years, very exciting to read indeed! The authors use state-of-the-art trophic marker analysis to show that not only the little available particulate organic carbon from surface sources but the 'old' inactive seep resources seem to (partially) support the high densities of these sponges, in addition to abundant bacterial symbionts inside the sponges. These were characterized by a suite of omics techniques

(which I am poorly qualified to evaluate) for both their composition and metabolic pathways, results that greatly help the interpretation of the possible and likely trophic pathways in the system. In addition to the figures, the high-resolution images are stunning and undoubtedly support the story. The manuscript shows that previously undiscovered habitat and food web types continue to be found in the Arctic at a time when discussions motivated by geopolitical and industrial interests regarding the future of the central Arctic Ocean are 'hot'. This work is without doubt suitable for the journal, the analyses are thorough, state of the art and professionally done, and the level of novelty is high. The writing needs some small language improvements here and the figures and tables would benefit from a few little tweaks (noted below). The number of replicates for the isotope analysis of pelagic POM, zooplankton, sea ice POM and fecal pellets are extremely low (and at least pelagic POM is not as hard to sample as the sponge community itself) which is regrettable, at least adding some comparative values from the literature can tell the readers if they are 'normal'. Densities of the sponges are given in relative terms, actual values seem to have been measured and should ideally be shown at least in the supplementary material. Small edits are listed below.

L23 delete 'a' before biomass

L126 grammar 'fluxes ... provide'

L26 add 'and' before compound-specific s.i. – and add 'of fatty acids'

L31 why unique to Arctic

L35 change are to have been (started in past, ongoing to present)

L48 clarify what 'process' means, clear of carbon/ particles?

L62 either is comprised of or comprising without of and is

L64 indicate if this includes pelagic and ice-associated primary production

L70 Which seamounts were investigated, so far one is named

L72 Big statement that lacks a reference!

L83 this appears to be a new species (in addition to genus)

L87 add 'on the massive *Geodia* sponges'

L87 is 'image estimates' necessary here? This should be in the methods

L88 replace starfishes by asteroids, consistent with ophiuroids (also at other places throughout the MS); fix sentence from 'as were' by giving actual densities of these echinoderms inside and outside of the seamounts and make a separate sentence.

L91 missing 'were' after Sponges

L92 What were the densities in these different areas? The supplementary information indicates they determined, but they are neither given here nor in the supplementary information.

L95 What kind of bivalves? They are given in L102, should rather be earlier to reduce redundancy.

L99 living colony of what? *Polybrachia*? Why colony? And what is the logical connection of 'no living colony' to organic matter content of the mats?

L225 spell out MOX and SOX

L250 should read summit (not plural), since only KM is named

L263ff What advective transports may be in the region that could complement vertical flux?

L269 the particulate matter feeding is sort of trivial to name here perhaps?

L333 Give us the animal tissue values please.

L339 remove one period

L346 add 'bacterial commonly being the main'

L391 the wording 'actively venting' is a bit misleading, because the study site was not actively venting as the authors point out.

L421 What type of box corer?

L435 Which layer(s) was the zooplankton collected at, and which species?

L447 How were the analyzed images chosen from the total taken?

L448 The average area imaged was 18.5 m², that seems like a lot. And how can the median be so much lower with 4.5 m (and is not given in m²)? The image scale in Fig 2 suggests the area is less than 18 m², or perhaps those images are cropped?

L457 cnidarians is missing an 'n'

L510 hyperbora should not have an 'n' at the end

Figure 3. Why is *G. parva* in yellow print while everything else is black? Unify.

Figure 4. remove some labels and thereby horizontal lines from y-axis to reduce clutter

Figure 5. The caption is not legible when printed in 100%

Table 1. No replication for zooplankton isotope samples, why? – I know from our own CAO sampling that even here it is unlikely that the material would have only been sufficient for a single

sample, especially since multinet samples were taken. And note what species are included and from which of the layers, all mixed up? But then they are not really discussed anywhere – can they be taken out? Missing from column header sediment 'layer', missing header for '10 m' in POM (sampling depth in water column?) and for zooplankton 'integration depth' if that is what is shown – they do not match with the multinet depths given in L435. Give consistent Latin names for macrofauna.

Supplementary material

L64 The Latin names Bryozoa, Hydrozoa are capitalized, but really they should be made consistent with the other terms, meaning bryozoans, hydrozoans. Please replace starfishes by asteroids. And which crustaceans were sampled here, specify.

L38, L43 and elsewhere, use unified abbreviation for hour.

Figure S3. Adding scales would be useful. And which sponges are we seeing here and from which seamount?

Figure S4. Is there space for this figure in the main body of the paper? It highlights the trophic structure nicely.

Table S3 add lines or some other separator between the types of biomarkers to help the reader's eyes.

Reviewer #3:

Remarks to the Author:

Nature Communications Report_Morganti et al.

Overview

The manuscript entitled "Giant sponge grounds of Arctic seamounts apparently associated with extinct seep life (Langseth Ridge, 87°N, 61°E)" by Morganti et al. is a very interesting one with some novel findings that enrich our knowledge about the biology and ecology of deep-sea sponge aggregations. Firstly, I would like to congratulate the authors for putting together this massive piece of work studying the biochemical/isotopic composition of *Geodia*, their microbial communities, the feeding biology of these holobionts as well as the biochemistry of potential food sources in areas for which we have very limited information on their structure and functioning. These are key steps to answer the critical question about how massive deep-sea sponges/aggregations thrive under (apparently) oligotrophic conditions. It looks, in overall, a well-organized study which has examined the appropriate points to give an answer to the important research question. I also think that this manuscript will raise further thinking about the biology and ecology of deep-sea fauna under relatively extreme conditions I feel that the authors provide some evidence supporting their suggestions/conclusions about the role of siboglinidae polychaetes' organic matter in the diet of sponges. In overall, however, I think that there are some major points that need to be addressed before any further consideration of this novel and interesting manuscript. Below I provide detailed comments but here are some key aspects:

1. My understanding is that the researchers have studied the stable isotope values ($\delta^{13}\text{C}$, $\delta^{15}\text{N}$) of both the sponge individuals and the potential food sources (DOM, POM etc.). I think that for this kind of manuscript is very important to provide some values for the relative contribution (%) of each food source in the diet of the sponges. This must be highlighted in the main manuscript. Currently this information is apparently missing, and this hinders the readers getting a better grasp. Once the role of food sources is given this will support the choice of the authors to highlight the role of extinct seep life in the title of their manuscript.

A couple of useful references are given below:

Parnell, A. C., Phillips, D. L., Bearhop, S., Semmens, B. X., Ward, E. J., Moore, J. W., et al. (2013). Bayesian stable isotope mixing models. *Environmetrics* 24, 387–399. doi: 10.1002/env.2221

Stock, B. C., Jackson, A. L., Ward, E. J., Parnell, A. C., Phillips, D. L., and Semmens, B. X. (2018). Analyzing mixing systems using a new generation of Bayesian tracer mixing models. *PeerJ* 6:e5096. doi: 10.7717/peerj.5096

2. It is true that the authors provide some lines of evidence about the role of siboglinidae polychaetes in the diet of sponges. What is not clear is how the particulate organic matter that is trapped underneath the sponges (i.e., found in the spicule mat) is mobilized and captured by the sponge holobiont. It seems that the authors have not suggested the relative feeding mechanism in their Discussion section; instead, they have highlighted (and they did very well) the massive filtering capacity of *Geodia*. I acknowledge the challenges associated with the description of the feeding mechanism capturing the underneath organic matter, but insightful suggestions must be given. Another point that I would like to bring to their attention about the mechanisms supporting the proliferation of these massive sponges under oligotrophic conditions is the possibility of sponges storing food sources supplied during episodic food-supply events. It would be good to see this in their Discussion section. Please see the publication below:

Maier et al. (2019): Survival under conditions of variable food availability: Resource utilization and storage in the cold-water coral *Lophelia pertusa*. *Limnol. Oceanogr.* 64, 2019, 1651–1671.

3. The authors need to account for the fact that the stable isotope values for key food sources (DIM, DOM, POM) have not been studied using samples collected at the study area; instead, they have used stable isotope values from the scientific literature (this is what they mention in the caption of their Figure 3). Using values from the literature is not necessarily bad but in the present case the spatial variability in the delta values of food sources could possibly introduce some bias in the interpretation/conclusions about the role of food sources in the *Geodia* diet. This is especially important for DOM and POM as *Geodia* is known to feed on both these sources. In addition, the authors need to consider and mention in the Discussion section of their manuscript further limitations in their study such as the minimum number of replicates used for most of the food sources (see Table 1 of the main manuscript).

4. The authors need to clarify that any contamination of sponge samples (analyzed for stable isotopes) from siboglinidae tubes material has been excluded. Such a contamination would have contributed to an overestimation of the role of siboglinidae organic matter in the diet of *Geodia*.
Specific Comments

Line 48: The text here does not sound very nice. The authors could use something such as "volume of water" instead "several hundred meters".

Line 51: Not sure if the term "opportunistic" is the best one that can be used to describe sponge feeding across the whole Phylum. I think that saying something like "...there is interspecific variability in sponge feeding strategies..." sounds more accurate.

Lines 52-54: Please provide a reference here showing that the sponges in the genera mentioned are HMA.

Lines 65-68: Based on the text given here, it seems that the authors have not collected samples from the water column to get the biochemical composition/isotope values of dissolved and particulate matter in the water column. If they have collected these samples, they really need to mention it here. If they have not collected these samples, then I am afraid that this is a major gap in this study.

Line 71: "...recalcitrant...". I assume the authors refer to refractory organic matter. Please use the term "refractory" for consistency with other parts of the manuscript.

Lines 70-72: I think here the authors should mention explicitly that they will be testing, among other, the role of siboglinidae tube detritus in the diet of *Geodia*.

Lines 34-72: I would suggest that the authors should avoid the use of impactful words (e.g., "substantial") when these are not accompanied by the relevant arithmetic values. For example, in the Introduction they mention "...we hypothesize that the sponges use recalcitrant organic matter as carbon, nitrogen and energy source, a substantial proportion of which was apparently produced during a phase of active venting of the seamounts several 1,000 years ago". If the authors can replace the term "substantial" by a value, then it is fine. Otherwise, they should just say "...source, a proportion of which...".

Lines 79-91: It is interesting to see this information about the fauna associated with the massive sponges, but I am not sure if it really adds to the main aims and objectives of the manuscript. The authors could place that to the Supplementary Information. This would give them the space to elaborate on more important aspects of their work [e.g., the role of (macro)habitat type, inclination, and depth in explaining sponge patterns of biomass and body size].
Line 92: If space permits, please mention the highest values of sponge density measured.

Line 111: "...the black tube detritus...". The authors should explain where they refer with these terms. Is this the siboglinidae tube detritus?

Line 119: "...with the small end...". Please rephrase e.g., "...the lower end...".

Line 123-126: The authors should have also a map (like the one in Figure 2a) showing the spatial distribution of sponge biomass (dry mass or carbon mass).

Line 125: Please leave a space between "average" and "1,213".

Lines 127-130: It is not clear how this information fits into the key aims and objectives of the manuscript e.g., the feeding biology / metabolism of these massive sponges.

Lines 133-135: It is not clear how the results about differences in the isotope values between adults and juveniles feed into the aims and objectives of the study. As mentioned above, the authors need to improve the connection between the rationale/introduction of their study (a null hypothesis in the introduction could perhaps help a lot) and the results shown.

Lines 137-138: Here the authors repeat the result about the statistically significant differences in $\delta^{13}\text{C}$ between *G. parva* adults and *G. parva* buds. Also please be consistent with the terminology used (see also my comment above). In the text it is mentioned as "buds" but in Table 1 is mentioned as "offspring". I think that neither of the two is a good choice; what about the term "juveniles"?

Line 141: "...other sample types...". I think it would be helpful for the reader if the authors had spelled out these samples e.g., sediments, suspended matter, "siboglinidae tubes".

Line 141: Since the authors refer to polychaete tubes (Fig. 3) I assume that they have analyzed the calcareous material that (usually) forms the tube of Serpulidae polychaetes. If this is the case, then I do not see why authors compare $\delta^{13}\text{C}$ isotope values from a mixture of organic/inorganic matter coming from sponges with the $\delta^{13}\text{C}$ values of inorganic matter from these tubes. I feel that this is not a meaningful comparison as usually there are large differences between the $\delta^{13}\text{C}$ values of organic matter and the $\delta^{13}\text{C}$ values of calcium carbonate material (e.g., see "Schlacher TA, Connolly RM (2014) Effects of acid treatment on carbon and nitrogen stable isotope ratios in ecological samples: a review and synthesis. *Methods Ecol Evol* 5:541-550. doi:10.1111/2041-210X.12183" and references therein). In the case that I have missed then the authors should increase the level of clarity in their text to help readers get a better grasp of the type of samples being analyzed each time.

Line 157: Please spell out (in the first point it is encountered) the measurement unit for age. I assume the "a" refers to plural of "annum".

Line 160: Data on the age of these massive cold-water sponges are precious and congratulations to the authors for providing this information.

Lines 166-178: I think that all this information is very helpful and need. However, I think that there is a key message missing. For example, the fact that "...Bacteria PLFAs were the most abundant (>60%)..." does it show that these sponges are HMA or do the authors want to emphasize on something else?

Line 190: I would suggest to the authors to use the term "refractory" (as they have done in parts of their work) across the whole manuscript.

Line 191: Please spell out the "PFAM" first and then you can use the abbreviation.

Lines 209-211: The authors could also consider also the manuscript from Van Duyl et al. 2008. van Duyl, F.C., Hegeman, J., Hoogstraten, A., Maier, C., 2008. Dissolved carbon fixation by sponge-microbe consortia of deep water coral mounds in the northeastern Atlantic Ocean. *Mar. Ecol. Prog. Ser.* 358, 137-150. <https://doi.org/10.3354/meps07370>

Line 216: The reference given here by the authors (i.e. Boetius, A. & Purser, 2017) comes from a scientific expeditions report. Are there any peer-reviewed publications that could support (further) the statement made here about the "...very low productivity and a sluggishness of currents...?"

Line 216: Please make a brief reference to the velocity of the currents helping in this way the readers to get a better grasp.

Lines 216-218: This is a key statement that the authors make, and they need to support it with references specific for their area of study.

Lines 226-229: I think that things here need a higher level of clarity. Specifically, "...detrital material...". Do the authors mean that the siboglinid tube worms were in detritus form? Please clarify.

Lines 229-231: This interesting hypothesis should be mentioned clearly in the "Introduction" of the paper to capture relatively quickly the interest of the readers.

Lines 229-231: Please clarify what you mean by "enriched". Do you refer to isotopically enriched material? I assume that this enriched material is also refractory.

Lines 229-231: I think that this sentence should be shorter and more impactful – the authors should consider excluding "on which they sit, including remnants of an extinct seep ecosystem...".

Line 254: I think that the word "what" is missing between "than" and "has".

Line 256: It is good that the authors consider the publications from Cathalot et al. 2015 and Kutti et al. 2013. Please mention, however, that that sponge biomass in Rockall Bank is relatively low (see the publications by Van Soest and Lavaleye, 2005; Van Soest et al. 2007). This means that the authors are comparing sponge oxygen consumption rates with oxygen consumption rates generated from other fauna (e.g., the fauna that dominates on dead cold-water coral framework, see Van Oevelen et al. 2009 for details). For cold-water coral and sponge carbon cycling the authors will find helpful the publication from De Clippele et al. 2021 in "Coral Reefs". All the references are given here to help you:

De Clippele, L.H., Rovelli, L., Ramiro-Sánchez, B., Kazanidis, G., Vad, J., Turner, S..., Roberts, J.M. (2021). Mapping cold-water coral biomass: an approach to derive ecosystem functions. *Coral Reefs*, 40, pp.215-231.

van Soest RWM, Lavaleye MSS (2005) Diversity and abundance of sponges in bathyal coral reefs of Rockall Bank, NE Atlantic, from boxcore samples. *Mar Biol Res* 1:338-349.

van SoestRWM, Cleary DFR, de Kluijver MJ, Lavaleye MSS, Maier C, van Duyl FC (2007) Sponge diversity and community composition in Irish bathyal coral reefs. *Contrib Zool* 76:121-142

Line 265: "...the carbon and nitrogen isotope values of the freshly deposited matter and suspended POM did not fit the values of the sponge community...". I looked at Fig.S4 but the authors have not clarified in that Figure which is the "freshly deposited matter" and which is the "suspended POM". In the S4 the authors only refer to "POM = particulate organic matter" without clarifying whether this is suspended POM or sedimented POM. They also do not provide any information for the status (fresh/decayed) of this POM. This is a key point and authors need to clarify things in the Figure.

Line 265: In addition, it seems that, in contrast to what the authors mention, the nitrogen isotope

values of POM match the ones from *Geodia parva*. The average $\delta^{15}\text{N}$ value of POM seems to be ~ 5 per mil. Considering an average food-web step of 3.4 per mil (Post, 2002) it seems that there is a pretty good match between the food source and the consumer. I acknowledge, though, that this 3.4 per mil step, is not globally applicable. I also acknowledge that there is a large distance between the $\delta^{13}\text{C}$ values of POM and the $\delta^{13}\text{C}$ values of sponges. I think that the authors need to reconsider and mention that the mismatch has mainly to do with $\delta^{13}\text{C}$.

Post DM. 2002. Using stable isotopes to estimate trophic position: models, methods, and assumptions. *Ecology* 83:703–18. doi:10.1890/0012-9658(2002)083[0703:USITET] 2.0.CO;2

Line 273: This is an important point for the authors' conclusions' so it should be supported by appropriate references.

Lines 275-278: The concentrations of DOM may be low in the study area, but sponges may have high efficiency in the retention and the assimilation of DOM into their biomass.

Line 277: It seems that DOM values in the study cited (Rossel et al. 2016) come from pore waters in sediments so I am not sure if they are comparable to the values measured by the authors. The authors can have a look in the DOM values given in Van Duyl et al. 2008. I give here the citation to help the authors.

van Duyl, F.C., Hegeman, J., Hoogstraten, A., Maier, C., 2008. Dissolved carbon fixation by sponge microbe consortia of deep water coral mounds in the northeastern Atlantic Ocean. *Mar. Ecol. Prog. Ser.* 358, 137–150. <https://doi.org/10.3354/meps07370>.

Line 286-287: It is not very clear what the authors want to say here about balance between DOC and DIC assimilation. Please improve the clarity of the sentence.

Line 297: "...is based on recalcitrant DOM as a carbon source...". What $\delta^{13}\text{C}$ values do the authors expect (approximately) for this refractory dissolved organic matter? Would these $\delta^{13}\text{C}$ values of DOM fit (in terms of a food source-consumer relationship) with the $\delta^{13}\text{C}$ values of *Geodia*?

Line 298: Were these sulfur-related genes expressed or not?

Line 300: "...tube matter...". Please mention clearly the composition of this tube matter. Is it mainly calcareous,...organic,...other...mixed?

Lines 331-332: What the authors mention here about the proximity between sponge and siboglinid isotope values is true but please keep in mind that the $\delta^{15}\text{N}$ values of the siboglinids (suggested food source for sponges) are higher than the $\delta^{15}\text{N}$ values of the sponges. It seems that this is an indication that siboglinid material has a rather minor contribution in the diet of *Geodia*. Please see my relevant comment in the start of the Review.

Line 342: "...amount of OM trapped...": The authors should give briefly the (assumed) composition of this OM e.g. "mixture of siboglinid tubes with refractory organic matter supplied from ocean surface" or something like that.

Like 343-344: This is also an important point where clarity should be higher. The authors have highlighted above that *Geodia* are filter feeders, and they filter huge amounts of seawater supporting their dietary requirements. However, it seems that the siboglinid organic matter lies underneath the sponges. How do sponges capture this "sedimented" organic matter? The authors need to make some insightful suggestions here about the food-capturing mechanism. Also, values are needed for the contribution of this food source to *Geodia*'s nutrition.

Lines 353-355: Since the authors feel that this is highly likely then it would be nice to see a scheme representing the fluxes of elements from one ecosystem component to another. This scheme could go even to the Supplementary Material.

Line 391: The chemosynthetic site is not active anymore, right?

Finally, I am not aware of the authors' plans for the use of their data, but I would suggest that

these data on deep-sea sponge density/biomass/distribution be deposited (after the publication of the manuscript) to online repositories such as the PANGAEA or the ICES Vulnerable Marine Ecosystems Data Base. This will help in the improved scientific understanding about the distribution of deep-sea sponge aggregations.

Figures:

Figure 1: The authors need to provide scale bars for all the images.

Figure 2a: The authors need to give measurement units (e.g., sponge individuals/m²).

Figure 3: The d¹⁵N isotope values of Bryozoa range a lot, more than 3.4 per mil. Any thoughts on that?

Tables:

Table 1: The isotope values of DOM are not given in the Table. My understanding is that authors have used values from the literature – these values should be given in that Table, accompanied by the relevant citation.

Table 1: I think that the stable isotope values for associated macro/megafauna (starfish etc.) do not add anything related to the aims & objectives the work. They should be placed in the Supplementary Material.

Supplementary Material:

Lines 60-61: Please explain/provide references why you used the Tukey test and not another for the pairwise comparisons.

Lines 64-65. The pooling of macrofauna across different trophic levels is not very helpful and does not give you much information.

In overall I think that the details given by the authors are adequate for the statistical analysis to be reproduced.

REVIEWER COMMENTS

Reviewer #1 (Remarks to the Author):

This beautiful study investigates the food sources, and the mechanisms of their assimilation by sponges from a fascinating location – the deep-sea Arctic seamounts. The fact that the authors collected samples from the seabed below the deep ice-covered waters is outstanding, given the technological effort behind the findings. This study uses a comprehensive bulk and compound-specific isotopic analysis, to show the contribution of DOM and DIC to the diet of sponge holobiont, and associates sponge diet with fossil seep detritus. Using omics, the authors highlight the contribution of microbial partners to sponge holobiont nutrition. Overall, this is a wonderful scientific effort, describing the functionality of hotspots of life in a rapidly-changing arctic environment.

My main concern is that the authors do not refer to the metagenomes when conducting the functional analyses of the community. In general, the information regarding the metagenomic sequencing appears to be lacking (how deeply were the metagenomes sequenced?). Did the authors try to bin the metagenome-assembled genomes, especially those of SAR202 and look at the expression profiles of specific microbes? Although the metatranscriptomes provide ample information regarding the microbial community function, metagenomics could also contribute to the understanding of the metabolic potential.

Answer 1: Dear Maxim Rubin-Blum, thank you very much for your constructive comments. We have sequenced metagenomes of the 13 sponge samples also used for metatranscriptomics (RNA/DNA co-extraction) and are currently preparing a companion paper on the binned metagenome-assembled genomes and their encoded metabolic pathways. As the current manuscript already covers a wide variety of ecological aspects, and metagenomic binning would add a whole new dimension to this study, we have decided to keep the two manuscripts separate.

Minor comments:

L186: Nitrosopumilus archaea – Candidatus prefix doesn't seem to be necessary, as Nitrosopumilus species have been cultivated and classified (https://www.microbiologyresearch.org/docserver/fulltext/ijsem/69/7/1892_ijsem003360.pdf?expires=1624340739&id=id&accname=guest&checksum=7204B3609C98A310994C2C51A9B5CCF5)

Answer 2: The Candidatus prefix was removed.

L203: Please consider the enzymes of the HP/HB cycle in archaeal Nitrosopumilus autotrophs.

Answer 3: We have found enzymes of this cycle in the metatranscriptome and summarized this in the discussion of carbon fixation pathways (please also see lines 226 -231).

L204: RuBisCO is not a gene. Please italicize *cbbLM*, *pmo*. How about *cbbM*?

Answer 4: This was changed to “RuBisCO-encoding genes, *pmo*...”

L207: There is no need to capitalize the cycle names.

Answer 5: The capital letters on the cycle names were replaced with lower case in the revised draft.

L225: ...methane- and sulfur-oxidizing symbionts. MOX and SOX are abbreviations not known to the readership (better acronyms are MOB and SOB, for methane- and sulfur-oxidizing bacteria).

Answer 6: MOX and SOX abbreviations were replaced, in the current draft are discussed as methane- and sulfur-oxidizing symbionts.

L231: Please rephrase “on which they sit”.

Answer 7: The “on which they sit” was removed (also please see Answer 35 to Reviewer 3). Throughout the manuscript this description for sponge position has been altered.

L321: “..including autotrophy”?

Answer 8: “Including autotrophic processes” was replaced by “including autotrophy” as suggested by the reviewer.

L339: Please remove a dot.

Answer 9: The dot was removed.

Maxim Rubin-Blum

Reviewer #2 (Remarks to the Author):

The authors present very spectacular findings of dense sponge aggregations on seamounts in the central Arctic Ocean where food supply from water column production and vertical flux is extremely low and not sufficient to support those sponge densities. Those of us who followed the media of this expedition have been waiting for manuscripts like this one for some years, very exciting to read indeed! The authors use state-of-the-art trophic marker analysis to show that not only the little available particulate organic carbon from surface sources but the ‘old’ inactive seep resources seem to (partially) support the high densities of these sponges, in addition to abundant bacterial symbionts inside the sponges. These were characterized by a suite of omics techniques (which I am poorly qualified to evaluate) for both their composition and metabolic pathways, results that greatly help the interpretation of the possible and likely trophic pathways in the system. In addition to the figures, the high-resolution images are stunning and undoubtedly support the story. The manuscript shows that previously undiscovered habitat and food web types continue to be found in the Arctic at a time when discussions motivated by geopolitical and industrial interests regarding the future of the central Arctic Ocean are ‘hot’. This work is without doubt suitable for the journal, the analyses are thorough, state of the art and professionally done, and the level of novelty is high. The writing needs some small language improvements here and the figures and tables would benefit from a few little tweaks (noted below). The number of replicates for the isotope analysis of pelagic POM, zooplankton, sea ice POM and fecal pellets are extremely low (and at least pelagic POM is not as hard to sample as the sponge community itself) which is regrettable, at least adding some comparative values from the literature can tell the readers if they are ‘normal’.

We thank the reviewer for his/her constructive comments. We have further improved the language with the support of a native English speaker, and we edited the grammar mistakes found by the reviewer as listed below. Additional details and info in the table and figure were added as suggested by the reviewer (please see our replies below each reviewer’s comment).

Answer 1: With regard to the number of replicates from other organic sources such as fecal pellets and ice algae, we fully understand the reviewer’s wish for more samples and data. However, sampling under ice, extracting plankton and particles from the oligotrophic seas is extremely time consuming. After we had discovered the sponge gardens, we revised the strategy of the mission to address the question of food sources and dimensions of the ecosystem, however, the *Polarstern* expedition PS101 was mostly targeted to the geological, geochemical and biological processes associated with the hydrothermal vents of Gakkel Ridge in another area some 6 hours steaming distance depending on ice conditions. Unfortunately, time was simply too limited to get more samples from pelagic POM, zooplankton and fecal pellets,

and samples collected assigned and shared between a range of other on board researchers investigating a diverse range of research questions.

It must be noted that there was enough material to analyze at least two replicates of each sample for the zooplankton (please see our answer 38), whereas for the fecal pellets and suspended POM, the collected material had to be pooled, as volumes were too low to allow division into two replicates. This low abundance is in line with our ecological assessment; the area is so oligotrophic that it is really difficult to sample sinking matter, thus supporting our hypothesis that the settling flux of nutrients is too low to provide a main food source for the sponges.

Following the reviewer's comment, we added in the discussion some comparative values from the bibliography in order to support our results (please see lines 386-388).

Densities of the sponges are given in relative terms, actual values seem to have been measured and should ideally be shown at least in the supplementary material.

Answer 2: The sponge density was calculated from across the three summits of the Langseth Ridge seamounts, also incorporating the densities observed across the saddle feature, and were given in the Figure 2 caption previously. In line with the reviewers comments, we now also present this density data within the main text.

Small edits are listed below.

L23 delete 'a' before biomass

Answer 3: "a" was deleted.

L126 grammar 'fluxes ... provide'

Answer 4: the grammar was corrected.

L26 add 'and' before compound-specific s.i. – and add 'of fatty acids'

Answer 5: the "and" before compound specific was already present. We added "of fatty acids" as suggested.

L31 why unique to Arctic

Answer 6: To the best of our knowledge, no other such dense sponge ecosystem has been discovered in the Central Arctic, therefore we think that the adjective "unique" is appropriate here. We have specified this in the revised MS (lines 451 - 456). Of course the Arctic region is much larger, and methane cold seeps have been discovered in the Barents Sea, also considered biological hotspots when compared to the surrounding seafloor (see the references cited in main text: Astrom, E. K. L. et al. Methane cold seeps as biological oases in the high-Arctic deep sea °. *Limnol. Oceanogr.* 63, 209–231

(2018), and Sen, A et al. Frenulate siboglinids at high Arctic methane seeps and insight into high latitude frenulate distribution. *Ecol. Evol.* 10, 1339–1351 (2020)). However, here we refer to ecosystems beyond 80°N, and so far this is a unique finding under permanent ice.

L35 change are to have been (started in past, ongoing to present)

Answer 7: Corrected.

L48 clarify what ‘process’ means, clear of carbon/ particles?

Answer 8: With this sentence we want to highlight the ability of sponges to filter large volume of water. Therefore, for clarity, we changed “process” with “filter” as well as modifying the surrounding text for clarity

L62 either is comprised of or comprising without of and is

Answer 9: The grammar was corrected.

L64 indicate if this includes pelagic and ice-associated primary production

Answer 10: This includes pelagic and ice algae primary production as the study site is permanently covered by sea-ice. The description of this material is now better presented in the revised text.

L70 Which seamounts were investigated, so far one is named

Answer 11: The seamounts description comes later in the text (see Material and Method). Now we have specified the number and names of the seamounts analyzed in this study. Generally throughout the text descriptions of peaks and regions of the Langseth Ridge have been improved for clarity.

L72 Big statement that lacks a reference!

Answer 12: This statement is supported by the radiocarbon results from this study. The radiocarbon analysis on the remnants of organic debris such as that of the siboglinid tubes provided that those tubes have been produced more than 1,000 years ago and is our own original finding (please see lines 179 - 184 and Table 1).

L83 this appears to be a new species (in addition to genus)

Answer 13: The reviewer is right, we modified the text as following “discovery of a new genus and species...”

L87 add ‘on the massive *Geodia* sponges’

Answer 14: The study Arctic sponge community is mostly composed by *Geodia parva*, *G. hentscheli* and *Stelletta raphidiophora*. Since it was not possible to distinguish the two taxa on images due to their similar morphology (as we specified in Material and Methods, section “Sponge abundance, biomass estimations and seafloor characterization”). We now write “on the massive Demosponges”, so all three main species are included.

L87 is ‘image estimates’ necessary here? This should be in the methods

Answer 15: The reviewer is right, the density estimates of all megabenthos were surveyed through image analysis as specified in the method section,

therefore the “image estimates” is redundant here and we removed it from the main text.

L88 replace starfishes by asteroids, consistent with ophiuroids (also at other places throughout the MS); fix sentence from ‘as were’ by giving actual densities of these echinoderms inside and outside of the seamounts and make a separate sentence.

Answer 16: “Starfish” was replaced with “asteroids” throughout the text and table. The density of these macrofauna was low compared to sponge density (lower than 0.2 ind/m²), however unusually high for this latitude of the Central Arctic. We added this information to the text.

The estimates outside the seamounts are from the cited work: Rybakova, E., Kremenetskaia, A., Vedenin, A., Boetius, A. & Gebruk, A. Deep-sea megabenthos communities of the Eurasian Central Arctic are influenced by ice-cover and sea624 ice algal falls. PLoS One 14, 1–27 (2019).

Further detailed studies are planned to be published in a forthcoming paper on mega -and macrofauna density and distribution observed at the wider Langseth Ridge, but are not the focus of this interdisciplinary study.

L91 missing ‘were’ after Sponges

Answer 17: “Were” was added

L92 What were the densities in these different areas? The supplementary information indicates they determined, but they are neither given here not in the supplementary information.

Answer 18: The densities are reported in the Figure 2 caption. Since we needed to cut substantially the length of the text in order to comply with the journal guidelines, we had decided to place the density estimates into the figure caption, they are however now given in the revised main text.

L95 What kind of bivalves? They are given in L102, should rather be earlier to reduce redundancy.

Answer 19: Lines 104 - 107 provide a general description of the substratum with which the sponges associate, while the following lines 107 - 116 provide specific details of the remnants of the organisms that were found and identified within the spicule mat, among them, the bivalve shells. For clarity we would prefer to keep that order in the manuscript.

L99 living colony of what? Polybrachia? Why colony? And what is the logical connection of ‘no living colony’ to organic matter content of the mats?

Answer 20: Yes, “living colony” refers to the occurrence of members of the Polybrachia and of mussels which typically form colonies at hydrothermal vents or Arctic cold seeps.

The logical connection of “no living colony” to the organic matter content of the mats is that most of it was composed of sponge spicule, bivalve shells and

Polybrachia tubes. The latter are made of chitin and proteinaceous matter (as explained in lines 108 - 110), therefore, the mat is organic matter-rich even though the individual organisms are long gone. This section has been revised in the text to better identify this material as having originated during the functioning of a now long dead extinct ecosystem.

L225 spell out MOX and SOX

Answer 21: They were replaced and spelled out as “methane- and sulfur-oxidizing symbionts”

L250 should read summit (not plural), since only KM is named

Answer 22: The reviewer is right, we corrected the grammar.

L263ff What advective transports may be in the region that could complement vertical flux?

Answer 23: As we detailed in lines 483 - 484, the oceanographic studies indicated no relevant advective processes to explain the observed biological densities.

L269 the particulate matter feeding is sort of trivial to name here perhaps?

Answer 24: The first sentence of the second section of the discussion was removed, so it starts now as “Recent investigations show that DOM assimilation may play the dominant role in sponge metabolism....”

L333 Give us the animal tissue values please.

Answer 25: The animal tissue values could not be analyzed in this study, because no living specimen of the Polybrachia or mussels were found. All the collected tubes and shells were empty. Therefore we could only analyze the bulk isotope composition of the tubes. Even though we would have liked to know the signature of the potentially thiotrophic or methanotrophic animals of the extinct seep, in the context of this study, the focus was on the tube material as potential food source. In the main text we now added an additional reference and the respective info of the ^{13}C values of siboglinid tube from Barents Sea.

[Astrom et al. 2019, Chemosynthesis influences food web and community structure in high-Arctic benthos, Mar. Ecol. Prog. Ser. 629: 19-42.]

L339 remove one period

Answer 26: One period was removed.

L346 add ‘bacterial commonly being the main’

Answer 27: “being” was added.

L391 the wording ‘actively venting’ is a bit misleading, because the study site was not actively venting as the authors point out.

Answer 28: Corrected to “Prior to our study, no such sponge ground nor an actively venting hydrothermal or cold seep community has been identified in the ice-covered Central Arctic $>80^\circ\text{N}$” We have at several points in the

revised manuscript better presented the fact that the surveyed section of the Langseth Ridge is no longer hydrothermally active.

L421 What type of box corer?

Answer 29: The box corer used was the USNEL box corer. This information was added in the material and method section.

L435 Which layer(s) was the zooplankton collected at, and which species?

Answer 30: The zooplankton was collected with vertical nets from the entire water column above the Ridge. This is better explained in this draft of the manuscript. The different Arctic and North-Atlantic copepods were the most abundant [Boetius, A. & Purser, A. The Expedition PS101 of the Research Vessel POLARSTERN to the Arctic Ocean in 2016 , Berichte zur Polar- und Meeresforschung = Reports on polar and marine research, Bremerhaven, Alfred Wegener Institute for Polar and Marine Research. (2017). doi:doi:10.2312/BzPM_0706_2017]

L447 How were the analyzed images chosen from the total taken?

Answer 31: The full set of sponge ground images from dive 169 were analysed, with every 10th image in the remaining data analysed. This information is better presented in the revised text.

L448 The average area imaged was 18.5 m², that seems like a lot. And how can the median be so much lower with 4.5 m (and is not given in m²)? The image scale in Fig 2 suggests the area is less than 18 m², or perhaps those images are cropped?

Answer 32: The slope of the seafloor in some areas resulted in the high mean average. This is now explained in the text. The median coverage estimation was much more representative of cover estimations for all the ecosystem categories discussed, other than the ridge flank images.

L457 cnidarians is missing an ‘n’

Answer 33: The grammar mistake was corrected.

L510 hyperbora should not have an ‘n’ at the end

Answer 34: The mistake was corrected.

Figure 3. Why is G. parva in yellow print while everything else is black? Unify.

Answer 35: G.parva in Figure 3 is written now in black

Figure 4. remove some labels and thereby horizontal lines from y-axis to reduce clutter

Answer 36: Figure 4 was modified as suggested by the reviewer.

Figure 5. The caption is not legible when printed in 100%

Answer 37: We increased the label size and added the names to the figure legend.

Table 1. No replication for zooplankton isotope samples, why? – I known

from our own CAO sampling that even here it is unlikely that the material would have only been sufficient for a single sample, especially since multinet samples were taken. And note what species are included and from which of the layers, all mixed up? But then they are not really discussed anywhere – can they be taken out? Missing from column header sediment ‘layer’, missing header for ‘10 m’ in POM (sampling depth in water column?) and for zooplankton ‘integration depth’ if that is what is shown – they do not match with the multinet depths given in L435. Give consistent Latin names for macrofauna.

Answer 38: The number of replicates in Table 1 indicates the number of biological replicates analyzed, which for the sponges indicate the number of separate individuals analyzed per species for each summit. For small animals such as zooplankton, hydrozoans and bryozoans and the polychaete tubes we could not analyze the single individuals or tubes due to limited amount of samples and the number of replicates (as shown in table 1) that we could obtain from each sampling station. For each station (two in KM and one station for NM and one for KM slope) the amount of matter collected was enough to analyze two replicates for each sample. The amount of zooplankton material that was analyzed in each replicate ranged from 0.98 to 3.39 mg DW.

With regard to the zooplankton sampling depth, we thank the reviewer to have noticed such discrepancy as there was an error in the reported depth in the main text. The zooplankton samples were collected with a side net attached to a multinet and they were collected as vertical hauls from the entire water column (so all mixed up), to assess any possible source of zooplankton carbon in our study. We edited the main text accordingly.

Sponges can ingest small zooplankton (see J. T. Hestetun, G. Tompkins-Macdonald, H. T. Rapp, A review of carnivorous sponges (Porifera: Cladorhizidae) from the Boreal North Atlantic and Arctic. *Zool. J. Linn. Soc.* 181, 1–69 (2017)), therefore we analyzed the zooplankton samples as possible food source for the study sponge species. The different $\delta^{13}\text{C}$ value of zooplankton as well as suspended POM and sponge tissue indicates that sponges do not rely on suspended live and detrital organic matter from the water column. The revised manuscript was edited to stress out this finding in the discussion (please see lines 294 - 298).

The Table 1 was edited and the column headers were added as suggested by the reviewer.

The Latin names were revised to be consistent throughout the manuscript, supplementary material and tables.

Supplementary material

L64 The Latin names Bryozoa, Hydrozoa are capitalized, but really they should be made consistent with the other terms, meaning bryozoans, hydrozoans. Please replace starfishes by asteroids. And which crustaceans were sampled here, specify.

Answer 39: The Latin names Bryozoa and Hydrozoa were replaced with bryozoans, hydrozoans as suggested by the reviewer, and starfishes was replaced with asteroids through the entire main text (see our previous answer 16) and supplementary material. The benthic crustaceans sampled belonged to the Order Amphipoda and Tanaidacea. This information is now provided in the caption of Table 1.

L38, L43 and elsewhere, use unified abbreviation for hour.

Answer 40: Hour was replaced and abbreviated through the supplementary material text as “hr”.

Figure S3. Adding scales would be useful. And which sponges are we seeing here and from which seamount?

Answer 41: As we explained in the material and method section, due to the similar morphology of the most prominent sponge species, it was not possible to distinguish them through images as agreed with the experts on this matter. Therefore we refrained from further specifying species.

Figure S4. Is there space for this figure in the main body of the paper? It highlights the trophic structure nicely.

Answer 42: The Figure S4 was added in the main text as suggested by the reviewer and it is now Figure 4.

Table S3 add lines or some other separator between the types of biomarkers to help the reader’s eyes.

Answer 43: Agreed, we added different colors as separators between the biomarkers types. We used the same colors as those used in Figure 5 (previous Figure 4) from the main text to be consistent.

Reviewer #3 (Remarks to the Author):
Nature Communications Report_Morganti et al.

Overview

The manuscript entitled “Giant sponge grounds of Arctic seamounts apparently associated with extinct seep life (Langseth Ridge, 87°N, 61°E)” by Morganti et al. is a very interesting one with some novel findings that enrich our knowledge about the biology and ecology of deep-sea sponge aggregations. Firstly, I would like to congratulate the authors for putting together this massive piece of work studying the biochemical/isotopic composition of *Geodia*, their microbial communities, the feeding biology of these holobionts as well as the biochemistry of potential food sources in areas for which we have very limited information on their structure and functioning. These are key steps to answer the critical question about how massive deep-sea sponges/aggregations thrive under (apparently) oligotrophic conditions. It looks, in overall, a well-organized study which has examined the appropriate points to give an answer to the important research question. I also think that this manuscript will raise further thinking about the biology and ecology of deep-sea fauna under relatively extreme conditions I feel that the authors provide some evidence supporting their suggestions/conclusions about the role of siboglinidae polychaetes’ organic matter in the diet of sponges. In overall, however, I think that there are some major points that need to be addressed before any further consideration of this novel and interesting manuscript. Below I provide detailed comments but here are some key aspects:

1. My understanding is that the researchers have studied the stable isotope values (d13C, d15N) of both the sponge individuals and the potential food sources (DOM, POM etc.). I think that for this kind of manuscript is very important to provide some values for the relative contribution (%) of each food source in the diet of the sponges. This must be highlighted in the main manuscript. Currently this information is apparently missing, and this hinders the readers getting a better grasp. Once the role of food sources is given this will support the choice of the authors to highlight the role of extinct seep life in the title of their manuscript. A couple of useful references are given below:

Parnell, A. C., Phillips, D. L., Bearhop, S., Semmens, B. X., Ward, E. J., Moore, J. W., et al. (2013). Bayesian stable isotope mixing models. *Environmetrics* 24, 387–399. doi: 10.1002/env.2221

Stock, B. C., Jackson, A. L., Ward, E. J., Parnell, A. C., Phillips, D. L., and Semmens, B. X. (2018). Analyzing mixing systems using a new generation of Bayesian tracer mixing models. *PeerJ* 6:e5096. doi: 10.7717/peerj.5096

Answer 1: We wish we could apply such a model, and have discussed this approach among the coauthors and tried various models, however, we came to the conclusion that with >four potential food sources and two stable isotopes, the model is underconstrained and multiple solutions are possible.

Just taking the data at face value, one might even conclude that the siboglinid tube accounts for close to 100 %, because its $\delta^{15}\text{N}$ is about 3-4 permille heavier. However, the sponge cannot access the tube material directly, it will be heavily reworked by microbes in the mat and the sponge symbionts.

In the manuscript we have now included a basic calculation to show a potential solution to the problem, but also explain that we cannot resolve the mixing of food sources, which may vary through the long lifetime of a sponge. Please see lines 368 - 383.

2. It is true that the authors provide some lines of evidence about the role of siboglinidae polychaetes in the diet of sponges. What is not clear is how the particulate organic matter that is trapped underneath the sponges (i.e., found in the spicule mat) is mobilized and captured by the sponge holobiont. It seems that the authors have not suggested the relative feeding mechanism in their Discussion section; instead, they have highlighted (and they did very well) the massive filtering capacity of *Geodia*. I acknowledge the challenges associated with the description of the feeding mechanism capturing the underneath organic matter, but insightful suggestions must be given. Another point that I would like to bring to their attention about the mechanisms supporting the proliferation of these massive sponges under oligotrophic conditions is the possibility of sponges storing food sources supplied during episodic food-supply events. It would be good to see this in their Discussion section. Please see the publication below:

Maier et al. (2019): Survival under conditions of variable food availability: Resource utilization and storage in the cold-water coral *Lophelia pertusa*. *Limnol. Oceanogr.* 64, 2019, 1651–1671.

Answer 2: We thank the reviewer for such constructive comment and for raising valuable points to improve the discussion.

Sponges within sponge grounds act as ecosystem engineers, the spicule mat creates a complex matrix where tubes are intermixed in the Langseth Ridge location, which also hydrodynamically trapping sinking POM to the seafloor. In other work on the same sponge community (Morganti et al. 2021), we described the frequent and abundant sponge trails visible within the seafloor images collected as indicative of the ability of sponge individuals to move/crawl on top of the spicule-tube mat. We hypothesize that this unique feature may be related to their feeding behavior: the massive sponge accumulation observed must rely, among other sources, on trapped material in

the spicule mat. It is likely that both the sedimentary microbes and the sponge microbiome are hydrolyzing the particulate material trapped, and that the hydrolysate enters the DOM pool, and is then taken up by the cells of the sponge holobiont.

We have expanded this explanation in the main text. Please see the last paragraph of the discussion lines 429 - 446.

We agree with the reviewer that episodic short intense export events of POM fluxes could be a recurrent feature in the study area, especially under the future climate change scenarios that foreseen sea-ice retreat, nonetheless, the total carbon flux is too small to explain the massive local sponge accumulation (Boetius et al. 2013; Wiedmann et al. 2020). Seasonal peaks of less than $10 \text{ mg C m}^{-2} \text{ d}^{-1}$ POM were reported at the Nansen Basin in July-August (Lalande et al. 2019). At the time of our sampling we did not observe algal falls at Langseth Ridge. Please see the revised discussion, lines 299 - 305.

[Lalande, C., Nothig, E.-M. & Fortier, L. Algal Export in the Arctic Ocean in Times of Global Warming. *Geophys. Res. Lett.* 46, 1–9 (2019).

Boetius A. et al., Export of algal biomass from the melting Arctic Sea Ice, *Science* 339, pp. 1430-1432 (2013). DOI: 10.1126/science.1231346

Wiedmann, I. et al. What Feeds the Benthos in the Arctic Basins? Assembling a Carbon Budget for the Deep Arctic Ocean. *Front. Mar. Sci.* 7, 224 (2020).]

3. The authors need to account for the fact that the stable isotope values for key food sources (DIM, DOM, POM) have not been studied using samples collected at the study area; instead, they have used stable isotope values from the scientific literature (this is what they mention in the caption of their Figure 3). Using values from the literature is not necessarily bad but in the present case the spatial variability in the delta values of food sources could possibly introduce some bias in the interpretation/conclusions about the role of food sources in the *Geodia* diet. This is especially important for DOM and POM as *Geodia* is known to feed on both these sources. In addition, the authors need to consider and mention in the Discussion section of their manuscript further limitations in their study such as the minimum number of replicates used for most of the food sources (see Table 1 of the main manuscript).

Answer 3: Yes, we combined DIM and DOM $\delta^{13}\text{C}$, $\delta^{15}\text{N}$ and $\Delta^{14}\text{C}$ isotope values from the literature from Central Arctic Ocean (Druffel et al. 2017; Thibodeau et al. 2017, Griffith et al. 2012 and Benner et al. 2005). Nonetheless, for the mat and the sponges, and a number of other sources we determined the $\delta^{13}\text{C}$, $\delta^{15}\text{N}$ and $\Delta^{14}\text{C}$ in this study. We are not aware of any

other such extensive data set for a deep-sea study, but agree with the referee, that further studies with in situ data would be helpful. Nonetheless, as sponges are very long lived, even that would leave questions to potential variations with time. Hence we believe that the combined field and literature data allow to support the hypotheses presented here.

Note that the reference values are from Arctic ocean including a nearby area, specifically from 88°8.96'N, 78°14.56'E in study Druffet et al. 2017.

In order to clarify that, we added short sentence in the manuscript under Material and method, section Radiocarbon dating (please see lines 602 - 604). And, as suggested below by the reviewer (see our Answer 56) such information was added in Table 1.

These considerations are now highlighted in the discussion section under “1. Carbon Demand”.

4. The authors need to clarify that any contamination of sponge samples (analyzed for stable isotopes) from siboglinidae tubes material has been excluded. Such a contamination would have contributed to an overestimation of the role of siboglinidae organic matter in the diet of *Geodia*.

Answer 4: We confirm that contamination of sponge samples from siboglinid tubes can be excluded, because the sponges used for this analysis were cleaned outside from any epibionts before the analysis, and only internal sponge tissue was used. For each sponge individual at least three pieces of sponge tissue were analyzed (not including the cortex, which is the most external part). We clarified it in the supplementary material and methods section under “Preparation for stable isotope analysis” how these samples were prepared (see lines 35 -36 in supplementary material) and we clarified that contamination from siboglinid tubes can be excluded in the main text Material and Method section, lines 568 – 570.

Specific Comments

Line 48: The text here does not sound very nice. The authors could use something such as “volume of water” instead “several hundred meters”.

Answer 5: The text was edited considerably and combining the suggestion from the reviewer 2 (please see our answer 8 to reviewer 2) hopefully now reads better.

Line 51: Not sure if the term “opportunistic” is the best one that can be used to describe sponge feeding across the whole Phylum. I think that saying something like “...there is interspecific variability in sponge feeding strategies...” sounds more accurate.

Answer 6: The text was edited and “opportunistic” was removed and the sentence now reads as “Sponges exploit different food sources”.

Lines 52-54: Please provide a reference here showing that the sponges in the genera mentioned are HMA.

Answer 7: The following reference was added.

The HMA-LMA Dichotomy Revisited: an Electron Microscopical Survey of 56 Sponge Species (2014) Volker Gloeckner, Markus Wehrl, Lucas Moitinho-Silva, Christine Gernert, Peter Schupp, Joseph R. Pawlik, Niels L. Lindquist, Dirk Erpenbeck, Gert Wörheide, and Ute Hentschel, *Bio. Bull* 227(1):78-88.doi: 10.1086/BBLv227n1p78

Lines 65-68: Based on the text given here, it seems that the authors have not collected samples from the water column to get the biochemical composition/isotope values of dissolved and particulate matter in the water column. If they have collected these samples, they really need to mention it here. If they have not collected these samples, then I am afraid that this is a major gap in this study.

Answer 8: We did collect samples from the water column for the isotope analysis of the particulate fractions (see-ice, suspended POM, fecal pellets and zooplankton), but unfortunately the dissolved fraction was not collected and therefore we used DOM values from the bibliography. Please note that the bulk seawater DOM is rather stable in composition (Rossel et al. 2020). The sentence was edited following the reviewer's suggestion (see lines 70 - 75) and we clarified in Table 1 the values that were taken from the bibliography. Please see also our reply to your third comment.

[Rossel Pamela E., Bienhold Christina, Hehemann Laura, Dittmar Thorsten, Boetius Antje (2020). Molecular Composition of Dissolved Organic Matter in Sediment Porewater of the Arctic Deep-Sea Observatory HAUSGARTEN (Fram Strait), *Frontiers in Marine Science*, 7: 428 DOI=10.3389/fmars.2020.00428]

Line 71: "...recalcitrant...". I assume the authors refer to refractory organic matter. Please use the term "refractory" for consistency with other parts of the manuscript.

Answer 9: The term "refractory" is now consistently used through the main text.

Lines 70-72: I think here the authors should mention explicitly that they will be testing, among other, the role of siboglinidae tube detritus in the diet of *Geodia*.

Answer 10: The sentence was modified following the reviewer's suggestion.

Lines 34-72: I would suggest that the authors should avoid the use of impactful words (e.g., "substantial") when these are not accompanied by the relevant arithmetic values. For example, in the Introduction they mention "...we hypothesize that the sponges use recalcitrant organic matter as carbon, nitrogen and energy source, a substantial proportion of which was apparently produced during a phase of active venting of the seamounts several

1,000 years ago”. If the authors can replace the term “substantial” by a value, then it is fine. Otherwise, they should just say “...source, a proportion of which...”.

Answer 11: Unfortunately, we could not separate and quantify the amount of siboglinid tubes and their fragments with other particles trapped into the spicule mat by weighing. The term “substantial” is from visually assessing the ratio, because the blackened siboglinid tubes in the box corer samples and seabed pictures are dominating the mat in terms of POC.

Lines 79-91: It is interesting to see this information about the fauna associated with the massive sponges, but I am not sure if it really adds to the main aims and objectives of the manuscript. The authors could place that to the Supplementary Information. This would give them the space to elaborate on more important aspects of their work [e.g., the role of (macro)habitat type, inclination, and depth in explaining sponge patterns of biomass and body size].

Answer 12: We have shortened the text on associated fauna, but find it essential to describe that the sponge grounds are a special ecosystem and biodiversity hotspot compared to the surrounding area (Wiedmann et al. 2020). This is important information for the context of the study.

Line 92: If space permits, please mention the highest values of sponge density measured.

Answer 13: The highest values of sponge density estimated on the flat summits were added.

Line 111: “...the black tube detritus...”. The authors should explain where they refer with these terms. Is this the siboglinidae tube detritus?

Answer 14: The reviewer is right, here we refer to the siboglinid tube detritus. The text was modified accordingly.

Line 119: “...with the small end...”. Please rephrase e.g., “...the lower end...”.

Answer 15: Corrected

Line 123-126: The authors should have also a map (like the one in Figure 2a) showing the spatial distribution of sponge biomass (dry mass or carbon mass).

Answer 16: We consider this advice to be useful, and a sponge biomass map would be a good addition, but as we computed biomasses based on average sponge numbers and sizes across areas (as derived from our limited OFOBS transects), we believe such an additional map, given our limited data, would basically replicate the zones already shown on Figure 2a. To add the information on biomass to this map and increase usefulness to the readers interested in this aspect of the work, we have placed carbon mass m^{-2} onto the figure key directly now, and reworded the figure legend.

Line 125: Please leave a space between “average” and “1,213”.

Answer 17: The space was added.

Lines 127-130: It is not clear how this information fits into the key aims and objectives of the manuscript e.g., the feeding biology / metabolism of these massive sponges.

Answer 18: This information adds valuable insight on the biology of this sponge community. The high abundance of observed juveniles indicates that the study sponge community is actively reproducing. This implies that they are allocating energy and food into reproduction investment, suggesting that they are not nutritionally constrained. This has been added into the discussion section 1. Carbon demand. Please see lines 272- 277.

Lines 133-135: It is not clear how the results about differences in the isotope values between adults and juveniles feed into the aims and objectives of the study. As mentioned above, the authors need to improve the connection between the rationale/introduction of their study (a null hypothesis in the introduction could perhaps help a lot) and the results shown.

Answer 19: We have provided and highlighted the key question we had in mind when conducting that assessment. It is important to assess the difference between young and old sponges to understand if the longevity is a factor in the isotope signature (please see lines 380 - 383). Also, we consider it an important observation that the sponges produce so many juveniles.

Lines 137-138: Here the authors repeat the result about the statistically significant differences in d13C between *G. parva* adults and *G. parva* buds. Also please be consistent with the terminology used (see also my comment above). In the text it is mentioned as “buds” but in Table 1 is mentioned as “offspring”. I think that neither of the two is a good choice; what about the term “juveniles”?

Answer 20: Through the main text, supplementary material, figures and tables the term “buds” was replaced with “juveniles”.

Line 141: “...other sample types...”. I think it would be helpful for the reader if the authors had spelled out these samples e.g., sediments, suspended matter, “siboglinidae tubes”.

Answer 21: The specific sample types were added.

Line 141: Since the authors refer to polychaete tubes (Fig. 3) I assume that they have analyzed the calcareous material that (usually) forms the tube of Serpulidae polychaetes. If this is the case, then I do not see why authors compare d13C isotope values from a mixture of organic/inorganic matter coming from sponges with the d13C values of inorganic matter from these tubes. I feel that this is not a meaningful comparison as usually there are large differences between the d13C values of organic matter and the d13C values of

calcium carbonate material (e.g., see “Schlacher TA, Connolly RM (2014) Effects of acid treatment on carbon and nitrogen stable isotope ratios in ecological samples: a review and synthesis. *Methods Ecol Evol* 5:541–550. doi:10.1111/2041-210X.12183” and references therein). In the case that I have missed then the authors should increase the level of clarity in their text to help readers get a better grasp of the type of samples being analyzed each time.

Answer 22: We analyzed the Serpulidae tubes, which were acidified as all other samples, in order to eliminate the inorganic carbon for the stable bulk $\delta^{13}\text{C}$ analysis. The samples preparation is described in details in the supplementary material.

The Serpulidae tube, composed mostly of calcium carbonate, also contains organic matter as glycoproteins synthesized by the organisms (Tanur et al. 2010; Vinn 2013). Therefore by acidifying the samples we aimed to remove the inorganic carbon. However, as best practice when analyzing samples with unknown and possibly high amount of inorganic carbon, we analyzed and compared the $\delta^{13}\text{C}$ of acidified and untreated samples (see the reference suggested by the reviewer, Schlacher et al. 2014, Synthesis and recommendation, point 8, page 548).

The $\delta^{13}\text{C}$ of untreated samples were much more enriched, showing a substantial effect of the inorganic carbon.

The results are shown here below for your interest

sample	$\delta^{13}\text{C}$ not acidified	$\delta^{13}\text{C}$ Acidified
	per mil	per mil
92#1B	0,21	-28
94#2B	0,05	-35
94#2B2	0,44	-22.8
94#3B	-1,65	-21.7
216#1B	0,44	-34.8

Note that this also accounts for the heavy local DIC values which we consider in the food model.

[Tanur AE, Gunari N, Sullan RM, Kavanagh CJ, Walker GC (2010) Insights into the composition, morphology, and formation of the calcareous shell of the serpulid *Hydroides dianthus*. *J Struct Biol* 169: 145–160.

Vinn O (2013) Occurrence, Formation and Function of Organic Sheets in the Mineral Tube Structures of Serpulidae (Polychaeta, Annelida). *PLoS ONE* 8(10): e75330. <https://doi.org/10.1371/journal.pone.0075330>]

Line 157: Please spell out (in the first point it is encountered) the measurement unit for age. I assume the “a” refers to plural of “annum”.

Answer 23: “a” is for annum (plurals annums). We spelled out at the first point is mentioned in the manuscript.

Line 160: Data on the age of these massive cold-water sponges are precious and congratulations to the authors for providing this information.

Answer 24: We thank the reviewer for the nice comment, he/she seems as enthusiastic as we are for this result!

Lines 166-178: I think that all this information is very helpful and need. However, I think that there is a key message missing. For example, the fact that “...Bacteria PLFAs were the most abundant (>60%)...” does it show that these sponges are HMA or do the authors want to emphasize on something else?

Answer 25: Yes, this is a further corroboration that the analyzed sponges are HMAs. Then, the relative contribution of each biomarker is important and we discussed it further as it provides additional information on the relative importance of food sources for these sponges (for instance, see the relatively low abundance of algae biomarkers).

Line 190: I would suggest to the authors to use the term “refractory” (as they have done in parts of their work) across the whole manuscript.

Answer 26: The term “recalcitrant” was removed and replaced by “refractory” through the main text.

Line 191: Please spell out the “PFAM” first and then you can use the abbreviation.

Answer 27: “PFAM” was spelled out the first time is mentioned in the manuscript.

Lines 209-211: The authors could also consider also the manuscript from Van Duyl et al. 2008. van Duyl, F.C., Hegeman, J., Hoogstraten, A., Maier, C., 2008. Dissolved carbon fixation by sponge–microbe consortia of deep water coral mounds in the northeastern Atlantic Ocean. *Mar. Ecol. Prog. Ser.* 358, 137–150. <https://doi.org/10.3354/meps07370>

Answer 28: We thank the reviewer for bring our attention to this work. This reference was added into the main text.

Line 216: The reference given here by the authors (i.e. Boetius, A. & Purser, 2017) comes from a scientific expeditions report. Are there any peer-reviewed publications that could support (further) the statement made here about the “...very low productivity and a sluggishness of currents...”?

Answer 29: The low productivity in the area has also been reported in Wiedmann et al. 2020; in Lalande et al.2019; the low current speed of the deep Atlantic waters has been reported in Woodgate 2013. But here, due to the limited number of cited references according to the journal guidelines and the fact that the cruise report is linked to data deposits, we prefer to cite the cruise report.

[Wiedmann, I. et al. What Feeds the Benthos in the Arctic Basins?

Assembling a Carbon Budget for the Deep Arctic Ocean. *Front. Mar. Sci.* 7, 224 (2020).

Lalande, C., Nothig, E.-M. & Fortier, L. Algal Export in the Arctic Ocean in Times of Global Warming. *Geophys. Res. Lett.* 46, 1–9 (2019).

Woodgate, R. (2013) Arctic Ocean Circulation: Going Around At the Top Of the World. *Nature Education Knowledge* 4(8):8]

Line 216: Please make a brief reference to the velocity of the currents helping in this way the readers to get a better grasp.

Answer 30: The reference to the current data was added.

Lines 216-218: This is a key statement that the authors make, and they need to support it with references specific for their area of study.

Answer 31: The following reference was added.

Woodgate, R. (2013) Arctic Ocean Circulation: Going Around At the Top Of the World. *Nature Education Knowledge* 4(8):8

Lines 226-229: I think that things here need a higher level of clarity. Specifically, “...detrital material...”. Do the authors mean that the siboglinid tube worms were in detritus form? Please clarify.

Answer 32: We have clarified this term by using “blackened siboglinid tube detrital material” as we have repeatedly used through the main text.

Lines 229-231: This interesting hypothesis should be mentioned clearly in the “Introduction” of the paper to capture relatively quickly the interest of the readers.

Answer 33: We further clarified our hypothesis and change the last sentence of the introduction. Please see lines 76 - 81.

Lines 229-231: Please clarify what you mean by “enriched”. Do you refer to isotopically enriched material? I assume that this enriched material is also refractory.

Answer 34: The term “enriched” was replaced with “refractory” for improving the clarity of the statement.

Lines 229-231: I think that this sentence should be shorter and more impactful – the authors should consider excluding “on which they sit, including remnants of an extinct seep ecosystem...”.

Answer 35: We do agree with the reviewer to shorten this sentence to increase the impact of the message we want to provide. Therefore we removed “on which they sit”. However, part of our main message is that most of such refractory material is composed by remnants of ancient seep biota, therefore we think it is important to keep the last part of the sentence that stress out such crucial information. We have reworded this portion of text to both improve the clarity and maintain this information.

Line 254: I think that the word “what” is missing between “than” and “has”.

Answer 36: The reviewer is right, “what” was added as he/she suggested.

Line 256: It is good that the authors consider the publications from Cathalot et al. 2015 and Kutti et al. 2013. Please mention, however, that that sponge biomass in Rockall Bank is relatively low (see the publications by Van Soest and Lavaleye, 2005; Van Soest et al. 2007). This means that the authors are comparing sponge oxygen consumption rates with oxygen consumption rates generated from other fauna (e.g., the fauna that dominates on dead cold-water coral framework, see Van Oevelen et al. 2009 for details). For cold-water coral and sponge carbon cycling the authors will find helpful the publication from De Clippele et al. 2021 in “Coral Reefs”. All the references are given here to help you:

De Clippele, L.H., Rovelli, L., Ramiro-Sánchez, B., Kazanidis, G., Vad, J., Turner, S..., Roberts, J.M. (2021). Mapping cold-water coral biomass: an approach to derive ecosystem functions. *Coral Reefs*, 40, pp.215-231.

van Soest RWM, Lavaleye MSS (2005) Diversity and abundance of sponges in bathyal coral reefs of Rockall Bank, NE Atlantic, from boxcore samples. *Mar Biol Res* 1:338–349.

van SoestRWM, Cleary DFR, de Kluijver MJ, Lavaleye MSS, Maier C, van Duyl FC (2007) Sponge diversity and community composition in Irish bathyal coral reefs. *Contrib Zool* 76:121–14

Answer 37: As suggested by the reviewer we specified in lines 265 - 267 that the sponge biomass is relatively low in Rockall Bank compared to the coral biomass. The biomass cited here 38 g C m⁻² represents the living biomass including sponges. We added such info also to improve clarity (please see line 266- 268).

We compared the sponge oxygen consumption rates and then the carbon demand of the study sponge community with: (i) other *Geodia* dominated deep-sea sponge communities from the Norwegian continental margins (Kutti et al. 2013; Cathalot et al. 2015 reported separately oxygen consumption by sponges and corals community see Table 5, page 10); (ii) sponge community dominated by *Thenea* spp. (Witte and Graf 1996); (iii) cold-water coral reefs including sponges and other organisms from Rockall Bank in which biomass is mostly compromised by corals (39%), sponges (19%) and infauna (12%) (van Oevelen et al. 2009) and from Northern Norway (Rovelli et al. 2015); (iv) and also cold-coral reefs from Norwegian continental margins excluding sponges (Cathalot et al. 2015).

We thus compared our estimates of oxygen and carbon demand with representative deep-sea megabenthos communities dominated by filter feeders: sponges dominated (sponge grounds), corals dominated and a mixture of corals, sponges and infauna.

We thank the reviewer to bring our attention to the recent publication from De Clippele et al. 2021. We now also use the published values in De Clippele et al. 2021 for comparison in this study (please see 286).

Line 265: "...the carbon and nitrogen isotope values of the freshly deposited matter and suspended POM did not fit the values of the sponge community...". I looked at Fig.S4 but the authors have not clarified in that Figure which is the "freshly deposited matter" and which is the "suspended POM". In the S4 the authors only refer to "POM = particulate organic matter" without clarifying whether this is suspended POM or sedimented POM. They also do not provide any information for the status (fresh/decayed) of this POM. This is a key point and authors need to clarify things in the Figure.

Answer 38: The term "freshly deposited matter" was removed and replaced for "sinking and suspended matter" followed by the analyzed samples for improving clarity. They are sea-ice material, zooplankton, fecal pellets and suspended POM. The Figure S4 legend was corrected as "POM susp= suspended particulate organic matter".

Line 265: In addition, it seems that, in contrast to what the authors mention, the nitrogen isotope values of POM match the ones from *Geodia parva*. The average $\delta^{15}\text{N}$ value of POM seems to be ~ 5 per mil. Considering an average food-web step of 3.4 per mil (Post, 2002) it seems that there is a pretty good match between the food source and the consumer. I acknowledge, though, that this 3.4 per mil step, is not globally applicable. I also acknowledge that there is a large distance between the $\delta^{13}\text{C}$ values of POM and the $\delta^{13}\text{C}$ values of sponges. I think that the authors need to reconsider and mention that the mismatch has mainly to do with $\delta^{13}\text{C}$.

Post DM. 2002. Using stable isotopes to estimate trophic position: models, methods, and assumptions. *Ecology* 83:703–18. doi:10.1890/0012-9658(2002)083[0703:USITET] 2.0.CO;2

Answer 39: We thank the reviewer for raising this point. We do agree that sponges may ingest suspended POM, but the $\delta^{13}\text{C}$ values measured in this study and the low availability of suspended POM in the study area show that sponges are using other local food sources. Such a finding is aligned with the low abundance of algae biomarkers observed in the PLFA profile of the analyzed sponges. However, we cannot exclude that during episodic POM inputs (e.g., seasonality) its relative contribution to the sponge diet may increase, as raised by the reviewer (Please see our answer 2).

In order to keep this part of the discussion short, and to acknowledge the different $\delta^{13}\text{C}$ values from sponges and suspended POM, we eliminated "and nitrogen" in the sentence so now it reads as: "In addition, the carbon isotope values of the ...".

Line 273: This is a important point for the authors conclusions' so it should be supported by appropriate references.

Answer 40: This sentence was rephrased and appropriate reference added.

Lines 275-278: The concentrations of DOM may be low in the study area, but sponges may have high efficiency in the retention and the assimilation of DOM into their biomass.

Answer 41: We agree with the reviewer's comment that DOM retention efficiency and assimilation might be high. But it must be noted that the average carbon uptake rates from the dissolved food source might be lower for HMAs compared to LMAs [$0.7 \mu\text{mol C}_{\text{DOM}} \text{mmol C}_{\text{sponge}}^{-1} \text{d}^{-1}$ for HMA *Geodia barretti*, $3.5 \mu\text{mol C}_{\text{DOM}} \text{mmol C}_{\text{sponge}}^{-1} \text{d}^{-1}$ LMA *Vazella pourtalesii*; Bart et al. 2021].

It must be noted also that sponges ingest DOC as a function of its ambient availability [Morganti et al. 2017; Mc Murray et al. 2016; Mueller et al. 2014] and a threshold of ambient DOC (about $80 \mu\text{mol/L}$) has been observed in different studies from different regions. Below that threshold sponges were observed to excrete DOC.

Since the studied sponges are HMAs, and the ambient DOC availability in the study site is around the $80 \mu\text{M}$ threshold, we point toward local sources as main food and energy sources for this giant sponge community in order to explain their presence in such oligotrophic environment.

[Bart, M. C. et al. Differential processing of dissolved and particulate organic matter by deep sea sponges and their microbial symbionts. *Sci. Rep.* 10, 1–13 (2020).

Morganti, T., Coma, R., Yahel, G. & Ribes, M. Trophic niche separation that facilitates coexistence of high and low microbial abundance sponges is revealed by in situ study of carbon and nitrogen fluxes. *Limnol. Oceanogr.* 62, 1963–83 (2017).

Mueller, B., J. de Goeij, M. Vermeij, Y. Mulders, E. van der Ent, M. Ribes, and F. C. Van Duyl. 2014. Natural diet of coral-excavating sponges consists mainly of dissolved organic carbon (DOC). *PLoS One* 9: e90152. doi:10.1371/journal.pone.0090152

McMurray, S. E., Z. I. Johnson, D. E. Hunt, J. R. Pawlik, and C. M. Finelli. 2016. Selective feeding by the giant barrel sponge enhances foraging efficiency. *Limnol. Oceanogr.* 61: 1–17. doi:10.1002/lno.10287]

Line 277: It seems that DOM values in the study cited (Rossel et al. 2016) come from pore waters in sediments so I am not sure if they are comparable to the values measured by the authors. The authors can have a look in the DOM values given in Van Duyl et al. 2008. I give here the citation to help the authors.

van Duyl, F.C., Hegeman, J., Hoogstraten, A., Maier, C., 2008. Dissolved carbon fixation by sponge microbe consortia of deep water coral mounds in the northeastern Atlantic Ocean. *Mar. Ecol. Prog. Ser.* 358, 137–150. <https://doi.org/10.3354/meps07370>.

Answer 42: The DOM values reported here are from bottom water. Please see that the cited paper Rossel et al. 2016 reports both DOM porewater, and DOM in bottom water of the Arctic (please see Rossel et al. 2016 in Table 1 the values are indicated with “Bw”). We have compared the Arctic and Atlantic data (from the reference suggested by the reviewer) and find that DOM concentration in Atlantic bottom water are within the range, slightly lower, than those reported by Rossel et al. 2016 from the Arctic water (51-73 μ M and 70-80 μ M, respectively).

Line 286-287: It is not very clear what the authors want to say here about balance between DOC and DIC assimilation. Please improve the clarity of the sentence.

Answer 43: The main message here is that the relatively young ^{14}C - age of the analyzed sponges suggests that they may obtain carbon from a relatively young C sources, such as through CO_2 fixation. This is further supported by the completed and expressed CO_2 fixation pathways detected in this study (see Table S5 for example).

The sentence was edited in order to improve the clarity.

Line 297: “...is based on recalcitrant DOM as a carbon source...”. What $\delta^{13}\text{C}$ values do the authors expect (approximately) for this refractory dissolved organic matter? Would these $\delta^{13}\text{C}$ values of DOM fit (in terms of a food source-consumer relationship) with the $\delta^{13}\text{C}$ values of *Geodia*?

Answer 44: There are two possible different sources of this DOM available at the study site:

- (1) The background deep-sea refractory DOM, for which we used the $\delta^{13}\text{C}$ value from the bibliography, which was -22.3 per mil from 300 m depth close to the study area (Druffel et. al. 2017). If ambient DOM would have been the main C source, it would require a fractionation from source-consumer of 4 per mil to account for the $\delta^{13}\text{C}$ sponge average of -18.4‰. Such fractionation is unlikely, as small shifts of 1-2 per mill are expected to separate an organism its food source (De Niro and Epstein 1978; Peterson and Fry 1987; Fry 1988).
- (2) Local refractory DOM, which is produced by the hydrolyzation of Siboglinid tubes by ambient or associated sponge microbes. The $\delta^{13}\text{C}$ of the Siboglinid tubes is -19.8 per mil. This $\delta^{13}\text{C}$ value fits with the expected fractionation of 1-2 per mill.

[DeNiro M.J. and Epstein S. Influence of diet on the distribution of carbon isotopes in animals. *Geochimica et Cosmochimica Acta*, 1978, Vol. 42, pp. 495-506.

Peterson B. J. and Fry B. Stable isotopes in Ecosystem studies. *Ann. Rev. Ecol. Syst.* 1987, 18:293-320.

Fry B. Food web structure on Georges Bank from stable C, N, and S isotopic compositions. *Limnology and Oceanography* 1988, 3, pp. 1182-1190.]

Line 298: Were these sulfur-related genes expressed or not?

Answer 45: yes, the sulfur-related genes were expressed. The verb “detected” was replaced with “expressed” to make this clearer to the reader.

Line 300: “...tube matter...”. Please mention clearly the composition of this tube matter. Is it mainly calcareous,...organic,...other...mixed?

Answer 46: The composition of the blackened tube matter is described in the result section (please see lines 107 - 110). However, in order to improve clarity, we added “siboglinid” to specify that we are referring to the organic tube matter detritus composed by siboglinid tubes, which are composed by chitin and proteinaceous matter.

Lines 331-332: What the authors mention here about the proximity between sponge and siboglinid isotope values is true but please keep in mind that the d15N values of the siboglinids (suggested food source for sponges) are higher than the d15N values of the sponges. It seems that this is an indication that siboglinids material have a rather minor contribution in the diet of *Geodia*. Please see my relevant comment in the start of the Review.

Answer 47: There are two important aspects to be considered: (i) The siboglinid tubes are not directly ingested by the sponges, but rather the tube material is hydrolyzed by bacteria, free-living and sponge symbionts - and serves as carbon and nitrogen source. (ii) The sponge microbiome composition and also its enzymatic capacity suggests a complex nitrogen cycling. Therefore the internal sponge nitrogen (re)-cycling determines a lower d15N (see our study lines 350 – 352, and Kahn et al.2018; Freeman et al. 2020 supplementary info).

However, the main point in this study is that a local source must be partially supplying the carbon demand of the sponges, given the discrepancy in biomass to the surrounding Arctic seafloor.

[Kahn, A. S., Chu, J. W. F. & Leys, S. P. Trophic ecology of glass sponge reefs in the Strait of Georgia, British Columbia. *Sci. Rep.* 8, 756 (2018).

Freeman et al. Microbial symbionts and ecological divergence of Caribbean, sponges: A new perspective on an ancient association *ISME*, 2020 <https://doi.org/10.1038/s41396-020-0625-3>]

Line 342: “...amount of OM trapped...”: The authors should give briefly the (assumed) composition of this OM e.g. “mixture of siboglinid tubes with refractory organic matter supplied from ocean surface” or something like that.

Answer 48: The composition of this OM was added as suggested.

Like 343-344: This is also an important point where clarity should be higher. The authors have highlighted above that *Geodia* are filter feeders, and they filter huge amounts of seawater supporting their dietary requirements.

However, it seems that the siboglinid organic matter lies underneath the sponges. How do sponges capture this “sedimented” organic matter? The authors need to need to make some insightful suggestions here about the food-capturing mechanism. Also, values are needed for the contribution of this food source to *Geodia*’s nutrition.

Answer 49: We kindly refer to our previous Answer 2 and we will briefly provide further explanation here below.

Briefly, our hypothesis is that the spicule mat acts as a sort of trap in which the remnants of past seep biota (e.g., siboginid tubes) and deposited POM (organic matter sinking from the ocean surface) lead to a local enrichment in carbon and nitrogen sources. Sponges tap into this refractory food source in which they crawl across the area covered by the blackened tubes. These are slowly hydrolyzed by bacteria, including the sponge-associated microbes which are specific for processing of refractory OM. The dissolved compounds (DOM, but also S^- and NH_4^+) can then be used by the sponge holobionts as food and energy source. This is supported by the meta-omics results, which showed the abundance of sponge-associated microbes involved in the OM recalcitrant degradation as well as expressed genes involved in recalcitrant OM degradation and sulfurized OM (please see second section of the discussion).

We observed abundant and frequent trails, left by the sponges, which are indicative that sponges may crawl/move on top of this spicule-tube mat. This original feature observed for the first time may be related to feeding behavior: by re-locating the sponges facilitate the decomposition of this OM and replace themselves in a better position to get such food source (Morganti et al. 2021).

In order to improve the clarity we rephrased this section of the manuscript. Please see lines 436 - 446.

[Morganti, T. M. et al. In situ observation of sponge trails suggests common sponge locomotion in the deep central Arctic. *Curr. Biol.* 31, R368–R370 (2021).]

Lines 353-355: Since the authors feel that this is highly likely then it would be nice to see a scheme representing the fluxes of elements from one ecosystem component to another. This scheme could go even to the Supplementary Material.

Answer 50: A scheme of the food sources and their relative importance in the sponge diet was added in the supplementary material, please see Figure S5.

Line 391: The chemosynthetic site is not active anymore, right?

Answer 51: Correct, the chemosynthetic site is not active. During the survey there was no evidence of hydrocarbon or methane emission, nor evidence of living benthic organisms typically associated with hydrocarbon or methane seeps. We eliminated the term “such” in the sentence as it might be confusing.

Throughout the revised text we have made it clearer that much of the material underlying the sponges originated in the now extinct vent ecosystem.

Finally, I am not aware of the authors plans for the use of their data, but I would suggest that these data on deep-sea sponge density/biomass/distribution be deposited (after the publication of the manuscript) to online repositories such as the PANGAEA or the ICES Vulnerable Marine Ecosystems Data Base. This will help in the improved scientific understanding about the distribution of deep-sea sponge aggregations.

Answer 52: It is a requirement for Nature Communication to make the data available in an open online repository. The biomass and density data set are already available at OSF.IO repository as detailed in the Data and Material availability section. We provide here the link for facility:

<https://osf.io/vcxye/>

The image data and cruise reports are publically available via the PANGAEA data archive. Links have been entered into the 'Data availability' section.

Figures:

Figure 1: The authors need to provide scale bars for all the images.

Answer 53: The scale bars were introduced for all the images.

Figure 2a: The authors need to give measurement units (e.g., sponge individuals/m²).

Answer 54: The densities of sponges are now given for all surveyed habitats, with the exception of the steep flanks of the Langseth Ridge, habitat category a. Here, the steep slope angle makes counting sponges difficult on the few shelves occasionally present in images. Additionally, the abundance of these shelves on unsurveyed sections of seamount flank cannot be extrapolated with confidence from the abundances of these structures in the areas surveyed. Certainly, density of sponges in these areas is low, more than an order of magnitude lower than in the other habitat areas, and of negligible significance in the current study. This has been made clearer in the revised text.

Figure 3: The d15N isotope values of Bryozoa range a lot, more than 3.4 per mil. Any thoughts on that?

Answer 55: The d15N is 9,5 +/- 1,5 per mil. The Figure 3 label was wrongly reported as these values represented bryozoan and hydrozoan skeleton pooled together. The label is now corrected and reported as in Table 1.

Tables:

Table 1: The isotope values of DOM are not given in the Table. My understanding is that authors have used values from the literature – these values should be given in that Table, accompanied by the relevant citation.

Answer 56: The reviewer is right, we used DOM values from the literature. This information is now added into Table 1. Please see also our reply to your comment 3.

Table 1: I think that the stable isotope values for associated macro/megafauna (starfish etc.) do not add anything related to the aims & objectives the work. They should be placed in the Supplementary Material.

Answer 57: We would like to present the full overview here. Such data are so rare for the Arctic, and as we have detected a previously unknown ecosystem we find it important to show the isotope values also for the non-sponge animals.

Supplementary Material:

Lines 60-61: Please explain/provide references why you used the Tukey test and not another for the pairwise comparisons.

Answer 58: A Tukey test was used because it allowed all possible pairs of group means (unplanned pairwise comparisons) and controlled for the Type I error. A short explanation with the corresponding reference is now provided. Please see line 68 in the supplementary material.

Lines 64-65. The pooling of macrofauna across different trophic levels is not very helpful and does not give you much information.

Answer 59: Due to limited sample size the different macrofauna samples (crustaceans, asteroids, bryozonas and hydrozoans) were grouped together for this study.

Our main objective here was not to characterize the entire food web, but rather distinguish the sponges from associated macrofauna from a statistical point of view. See Table S2.

In overall I think that the details given by the authors are adequate for the statistical analysis to be reproduced.

Reviewers' Comments:

Reviewer #1:

Remarks to the Author:

Congratulations on the spectacular findings and the state-of-the-art science. The authors have satisfactorily addressed my comments, and I find the manuscript suitable for publication in the current form.

Maxim

Reviewer #2:

Remarks to the Author:

The authors prepared a thorough and satisfactory reply letter and their revision of the article reflects what they wrote in the letter. The revisions have enhanced the clarity of the presentation and interpretation of the undoubtedly noteworthy results. Green light for publication from my side.

Reviewer #3:

Remarks to the Author:

Morganti et al.

Dear Editor / Dear Authors,

I would like to congratulate the authors for the work they did to address my comments. Their efforts are much appreciated. I suggest the acceptance of publication of this manuscript. There are only a few minor points that I would like to raise. I feel these points can improve the already very high quality of the work. Please see below:

1. I acknowledge and agree with the suggestion made by the authors about how the sedimented/trapped organic matter becomes accessible to *Geodia*. I think, however, that there is also another (relatively simple) explanation about how sedimented organic matter becomes available to suspension/filter feeding sponges. This possible explanation has to do with the resuspension (e.g. through internal waves, benthic storms) of the sedimented/trapped organic matter. The settled material is suspended by physical phenomena and as it settles back on the seafloor becomes available to the sponges. This possible explanation will be helpful to address any questions that may be raised by the scientific community around the feeding mechanism. The feeding mechanism is a key point in the publication, and I feel that it needs to be supported even further.
2. Lines 436-446 (in the revised manuscript). I would suggest that the authors support their suggestions about the feeding mechanism (e.g. role hydrolysis) with a couple of appropriate references; currently this section has no references at all. For example, the authors should provide references about hydrolysis mediated by symbiotic microbes in marine invertebrate/sponges (references from shallow waters organisms could also be used).
3. Line 299: logistical reasons. Please add reference to the relevant Table with the number of sample replicates.

Rebuttal letter for the reviewers

*We thank the reviewers for their constructive and positive comments.
Here below please find our reply to reviewer #3 in italics*

REVIEWERS' COMMENTS

Reviewer #1 (Remarks to the Author):

Congratulations on the spectacular findings and the state-of-the-art science. The authors have satisfactorily addressed my comments, and I find the manuscript suitable for publication in the current form.

Maxim

Authors: Thank you very much for your constructive comments which have substantially improved the manuscript.

Reviewer #2 (Remarks to the Author):

The authors prepared a thorough and satisfactory reply letter and their revision of the article reflects what they wrote in the letter. The revisions have enhanced the clarity of the presentation and interpretation of the undoubtedly noteworthy results. Green light for publication from my side.

Authors: Thank you very much for your constructive comments which have substantially improved the manuscript.

Reviewer #3 (Remarks to the Author):

Morganti et al.

Dear Editor / Dear Authors,

I would like to congratulate the authors for the work they did to address my comments. Their efforts are much appreciated. I suggest the acceptance of publication of this manuscript. There are only a few minor points that I would like to raise. I feel these points can improve the already very high quality of the work. Please see below:

1. I acknowledge and agree with the suggestion made by the authors about how the sedimented/trapped organic matter becomes accessible to Geodia. I think, however, that there is also another (relatively simple) explanation about how sedimented organic matter becomes available to suspension/filter feeding sponges. This possible explanation has to do with the resuspension (e.g. through internal waves, benthic storms) of the sedimented/trapped organic matter. The settled material is suspended by physical phenomena and as it settles back on the seafloor becomes available to the sponges. This possible explanation will be helpful to address any questions that may be raised by the scientific community around the feeding mechanism. The feeding mechanism is a key point in the publication, and I feel that it needs to be supported even further.

Answer 1: We agree that a possible mechanism that allows sponges to tap into the OM trapped in the complex spicule mat matrix might be due to resuspension by currents such as internal tides, as inferred in other studies (please see Rice et al. 1990 and Kahn et al. 2018, Roberts et al. 2018). However, it must be noted that such a resuspension of bottom sediments requires rather strong

currents as known to occur in some continental slope/shelf-edges (e.g., current speed during the flood tides reached up to 92 cm s^{-1} in Fraser Ridge, glass sponge reef – Leys et al. 2011)(Dickson & McCave 1986, Thorpe & White 1988). In the studied area of an open ocean seamount, we found no evidence of such strong bottom currents, but we detected movement of sponges across the seafloor that we relate to feeding behavior. To discuss all options, we have added a respective sentence in the discussion section. Please see lines 445-449.

[Rice, A. L., Thurston, M. H. & New, A. L. Dense aggregations of a hexactinellid sponge, *Phoronema carpenteri*, in the Porcupine Seabight (northeast Atlantic Ocean), and possible causes. *Prog. Oceanogr.* 24, 179–196 (1990).

Kahn, A. S., Chu, J. W. F. & Leys, S. P. Trophic ecology of glass sponge reefs in the Strait of Georgia, British Columbia. *Sci. Rep.* 8, 756 (2018).

Roberts, E. M. et al. Oceanographic setting and short-timescale environmental variability at an Arctic seamount sponge ground. *Deep. Res. Part I Oceanogr. Res. Pap.* 138, 98–113 (2018).

Leys, S. P., G. Yahel, M. A. Reidenbach, V. Tunnicliffe, U. Shavit, and H. M. Reiswig. 2011. The sponge pump: The role of current induced flow in the design of the sponge body plan. *PLoS One* 6: e27787. doi:10.1371/journal.pone.0027787

Dickson RR, McCave IN (1986) Nepheloid layers on the continental slope west of Porcupine Bank. *Deep-Sea Res A, Oceanogr Res Pap* 33: 791–818

Thorpe SA, White M (1988) A deep intermediate nepheloid layer. *Deep-Sea Res A, Oceanogr Res Pap* 35: 1665–1671]

2. Lines 436-446 (in the revised manuscript). I would suggest that the authors support their suggestions about the feeding mechanism (e.g. role hydrolysis) with a couple of appropriate references; currently this section has no references at all. For example, the authors should provide references about hydrolysis mediated by symbiotic microbes in marine invertebrate/sponges (references from shallow waters organisms could also be used).

Answer 2: Thank you for this suggestion, the following references were added:

Bayer, K., Jahn, M. T., Slaby, B. M., Moitinho-Silva, L. & Hentschel, U. Marine Sponges as *Chloroflexi* Hot Spots: Genomic Insights and High-Resolution Visualization of an Abundant and Diverse Symbiotic Clade. *mSystems* 3, e00150-18 (2018).

Kamke, J. et al. Single-cell genomics reveals complex carbohydrate degradation patterns in poribacterial symbionts of marine sponges. *ISME J.* 7, 2287–2300 (2013).

3. Line 299: logistical reasons. Please add reference to the relevant Table with the number of sample replicates.

Answer 3: The reference to Table 1 and the number of sample replicates are added in the main text.